# Multi-color dSTORM microscopy in *Hormad1*[-/-] spermatocytes reveals alterations in meiotic recombination intermediates and synaptonemal complex structure

Lieke Koornneef[1,2], Johan A. Slotman[3], Esther Sleddens-Linkels[1], Wiggert A. van Cappellen[3], Marco Barchi[4], Attila Tóth[5], Joost Gribnau[1,2], Adriaan B. Houtsmuller[3], Willy M. Baarends[1]*

1 Department of Developmental Biology, Erasmus MC, Rotterdam, The Netherlands, 2 Oncode Institute, Erasmus MC, Rotterdam, The Netherlands, 3 Department of Pathology, Erasmus Optical Imaging Center, Erasmus MC, Rotterdam, The Netherlands, 4 Department of Biomedicine and Prevention, Faculty of Medicine, University of Rome Tor Vergata, Rome, Italy, 5 Institute of Physiological Chemistry, Faculty of Medicine at the TU Dresden, Dresden, Germany

* w.baarends@erasmusmc.nl

**Data Availability Statement:** The microscopic data generated in this study (dSTORM and STED) have

## Abstract

Recombinases RAD51 and its meiosis-specific paralog DMC1 accumulate on single-stranded DNA (ssDNA) of programmed DNA double strand breaks (DSBs) in meiosis. Here we used three-color dSTORM microscopy, and a mouse model with severe defects in meiotic DSB formation and synapsis (*Hormad1*[-/-]) to obtain more insight in the recombinase accumulation patterns in relation to repair progression. First, we used the known reduction in meiotic DSB frequency in *Hormad1*[-/-] spermatocytes to be able to conclude that the RAD51/DMC1 nanofoci that preferentially localize at distances of ~300 nm form within a single DSB site, whereas a second preferred distance of ~900 nm, observed only in wild type, represents inter-DSB distance. Next, we asked whether the proposed role of HORMAD1 in repair inhibition affects the RAD51/DMC1 accumulation patterns. We observed that the two most frequent recombinase configurations (1 DMC1 and 1 RAD51 nanofocus (D1R1), and D2R1) display coupled frequency dynamics over time in wild type, but were constant in the *Hormad1*[-/-] model, indicating that the lifetime of these intermediates was altered. Recombinase nanofoci were also smaller in *Hormad1*[-/-] spermatocytes, consistent with changes in ssDNA length or protein accumulation. Furthermore, we established that upon synapsis, recombinase nanofoci localized closer to the synaptonemal complex (SYCP3), in both wild type and *Hormad1*[-/-] spermatocytes. Finally, the data also revealed a hitherto unknown function of HORMAD1 in inhibiting coil formation in the synaptonemal complex. SPO11 plays a similar but weaker role in coiling and SYCP1 had the opposite effect. Using this large super-resolution dataset, we propose models with the D1R1 configuration representing one DSB end containing recombinases, and the other end bound by other ssDNA binding proteins, or both ends loaded by the two recombinases, but in below-resolution proximity. This may then often evolve into D2R1, then D1R2, and finally back to D1R1, when DNA synthesis has commenced.

been deposited in the BioImage Archive (http://www.ebi.ac.uk/bioimage-archive) under accession number S-BIAD423. Source data are provided with this paper.

**Funding:** LK and JG are supported by the Oncode Institute: https://www.oncode.nl/about/about-oncode. The funders had no role in study design, data collection and analysis, decision to publish, or preparation of the manuscript.

**Competing interests:** The authors have declared that no competing interests exist.

## Author summary

In order to correctly pair homologous chromosomes in the first meiotic prophase, repair of programmed double strand breaks (DSBs) is essential. By unravelling molecular details of the protein assemblies at single DSBs, using super-resolution microscopy, we aim to understand the dynamics of repair intermediates and their functions. We investigated the localization of the two recombinases RAD51 and DMC1 in wild type and HORMAD1-deficient cells. HORMAD1 is involved in multiple aspects of homologous chromosome association: it regulates formation and repair of DSBs, and it stimulates formation of the synaptonemal complex (SC), the macromolecular protein assembly that connects paired chromosomes. RAD51 and DMC1 enable chromosome pairing by promoting the invasions of the intact chromatids by single-stranded DNA ends that result from DSBs. We found that in absence of HORMAD1, RAD51 and DMC1 showed small but significant morphological and positional changes, combined with altered kinetics of specific RAD51/DMC1 configurations. We also determined that there is a generally preferred distance of ~900 nm between meiotic DSBs along the SC. Finally, we observed changes in the structure of the SC in *Hormad1*[-/-] spermatocytes. This study contributes to a better understanding of the molecular details of meiotic homologous recombination and the role of HORMAD1 in meiotic prophase.

## Introduction

During meiosis, programmed DNA double strand breaks (DSBs) are induced to serve as starting point in the alignment of homologous chromosomes. Repair of DSBs proceeds via a specialized form of homologous recombination repair. This is essential for homologous chromosome pairing, and results in physical connections between the homologous chromosomes. These, so-called chiasmata, are crucial for the first meiotic division. At the onset of meiotic prophase, and after induction of the DSBs, the single-stranded DNA (ssDNA) ends that are formed after end resection of meiotic DSBs, are bound by RPA and other meiosis-specific proteins (reviewed by [1,2]). Subsequently, these proteins are replaced by the recombinase RAD51 and its meiosis-specific paralog DMC1 to perform strand invasions into the homolog or the sister chromatid (reviewed by [3,4]). Genetic evidence indicates that both RAD51 and DMC1 are required for proper repair of meiotic DSBs in species ranging from yeast [5,6], plants [7,8], to mammals [9–11], although they have different functions. If DSBs form in somatic cells, RAD51 also functions in the repair of DSBs by homologous recombination, which involves preferential use of the sister chromatids as targets of strand invasions [12]. However, to achieve homologous chromosome pairing and to generate crossovers during mammalian meiosis, it is essential that strand invasions occur (also) into the homolog. A process called interhomolog bias is thought to be involved in mediating this switch in strand invasion target. The interhomolog bias is well studied in *S. cerevisiae*, where in the absence of the major players of the interhomolog bias, Red1, Hop1 and Mek1, the sister chromatid is the preferred strand invasion target for repair [13–17]. Currently, it is thought that these proteins support DMC1 to actively engage in strand invasion with the homolog, and simultaneously suppress RAD51 enzymatic activity [17–20]. This concept is supported by data from biochemical and evolutionary analyses which indicate that there are a few amino acid differences between RAD51 and DMC1 that allow DMC1 to form stable connections in spite of differences between homologous chromosomes in the exact nucleotide sequences, compared to the strict homology required for RAD51-mediated strand invasion [21–24]. These proposed

different contributions of RAD51 and DMC1 to repair are expected to be reflected in differences in loading on the ssDNA ends.

The reported meiotic localization patterns of RAD51 and DMC1 vary among model organisms. In mouse spermatocytes and oocytes, immunocytochemical analyses indicate that both recombinases largely overlap, but in *A. thaliana* meiocytes, RAD51 and DMC1 form single or heteromeric doublets, leading to the hypothesis that RAD51 and DMC1 occupy opposing ends of the DSBs [25–27]. However, results from a super-resolution study in *S. cerevisiae* suggested that RAD51 and DMC1 co-assemble on both ends of the DSB, visible as paired co-foci containing both RAD51 and DMC1 at approximately 400 nm distance from each other [18]. In addition, recent structured illumination microscopy (SIM) data on RAD51 localization in *C. elegans*, revealed paired RAD51 foci occurrence (*C. elegans* lacks a DMC1 homolog, but RAD51 has DMC1-like properties [24,28,29]). In contrast, in the protist Tetrahymena, DMC1 foci are clearly present, but RAD51 foci are not visible, although both proteins are required for meiosis [30]. In contradiction to yeast where paired co-foci were observed [18], two-color dSTORM (direct stochastic optical reconstruction microscopy) combined with SIM on mouse spermatocytes showed that recombination foci with one nanofocus of RAD51 and one of DMC1, called D1R1, are the main configuration [18,31]. Both the yeast and mouse study also revealed large variability in accumulation patterns [18,31]. In contrast, ChIP-seq of RAD51 and DMC1 in mice showed a more uniform pattern, with DMC1 located at the 3' ends of the ssDNA and RAD51 assembly closer towards the double stranded DNA (dsDNA) [32]. These differences may be due to the fact that the ChIP-seq readout is a sum of multiple individual assemblies but includes genomic localization, while super-resolution microscopy allows analysis at individual sites but lacks precise information on genomic localization (reviewed by [3]). To understand how the interplay between RAD51 and DMC1 accomplishes homology recognition, and contributes to repair, further functional analyses are required. In this paper, we have addressed this question by comparing the localization of RAD51 and DMC1 on DSBs between wild type and HORMAD1-deficient mouse spermatocytes.

Deficiency in *Hormad1* leads to infertility in both sexes, reduction of recombination foci numbers of RAD51, DMC1, and RPA, and meiotic arrest at an early pachytene-like stage, but with incomplete chromosome synapsis [33–35]. This so-called Stage IV or pachytene arrest, is most likely caused by a recombination-dependent arrest mechanism, followed by apoptotic elimination triggered by failure of XY body formation [36–39]. HORMAD1 is involved in several distinct processes during meiosis. First, HORMAD1 is associated with the chromosome axis where it may enhance the formation/stability of the synaptonemal complex (SC), a protein complex that consists of axial/lateral, transverse and central elements [34,40,41]. Second, HORMAD1 is also involved in the formation of DSBs by enabling enhanced axial accumulation of the DSB machinery consisting of IHO1, REC114, MEI4, MEI1, the recently discovered protein ANKRD31 [42–45], and possibly other unknown components. Third, HORMAD1 enables efficient accumulation and activation of the DNA damage response kinase ATR on unsynapsed axes, which is particularly critical for a female-specific meiotic checkpoint [34,46,47]. Fourth, HORMAD1 is a mammalian homolog of Hop1, one of the proteins involved in the interhomolog bias in budding yeast [48]. A second Hop1 homolog in mice is HORMAD2 [41]. If and how HORMAD1 is involved in the interhomolog bias, remains unknown. In the last decade, several studies have revealed an additional role for HORMAD1 and HORMAD2 in meiotic DSB repair [35,49,50]. In a SPO11-deficient background, protein markers of ssDNA turned over faster in the absence of HORMADs as compared to wild type, suggesting a role for HORMADs in DSB repair pathway choice or in inhibiting meiotic DSB repair in general [49,50]. Although it should be noted that the DSBs analyzed were radiation-induced, both RAD51 and DMC1 accumulate at such DSBs, they (re)localize to the SC, and contribute to homology recognition [50]. Also, Cisplatin-induced DSBs were shown to contribute to homologous chromosome synapsis in a hybrid sterility mouse model [51].

The above-proposed functions of HORMAD1 in DSB induction and repair prompted us to ask whether in the absence of HORMAD1 the accumulation patterns of recombinases at meiotic DSBs would alter. In particular, we felt that the strong reduction in the number of DSBs could help to delineate how recombinase nanofoci relate to individual DSB sites. In addition, we wanted to assess whether the lifetime of certain configurations would be altered due to the loss of a possible repair-inhibiting function. Furthermore, we aimed to obtain progress in interpreting super-resolution images of recombinase accumulation in the context of SC components, using three-color dSTORM to allow more precise analyses of RAD51 and DMC1 localization patterns relative to the lateral and axial elements of the SC. In addition, using this three-color approach we could generate an independent dataset with other fluorophore combinations that allowed testing the robustness of our previous dataset, and extending the nano-focus feature set. Our results show that each $D_x R_y$ recombinase configuration represents one DSB site, and that HORMAD1 influences the lifetime of RAD51 and DMC1 assemblies as well as their structural properties. Unexpectedly, we also identified a function of HORMAD1 in inhibiting coiling of the SC, and provide evidence that coiling can be affected by altered SC structure as well as by meiotic DSBs. Based on these findings and other data, we propose test-able models of recombinase accumulation to functionally interpret the microscopic images.

## Results

### HORMAD1 localization on meiotic chromosome axes

First, we determined the localization of HORMAD1 along the axial and lateral elements of the SC during meiotic prophase in spermatocytes. HORMAD1 was observed along the axial elements and depleted from synapsed regions (Fig 1A (zygotene), 1E (early pachytene)), as previously described [41]. We used dSTORM imaging on mouse spermatocytes which showed a more detailed structure of the HORMAD1 accumulation on unsynapsed regions in zygotene, and revealed HORMAD1 localizing as double axes (Fig 1B and 1C). We confirmed this observation by measuring intensity profiles along a line perpendicular to a manually-drawn axis-centered line clearly showing the distribution of HORMAD1 in two axes adjacent to SYCP3 (Fig 1D, for details see Material and Methods, S1 and S2 Figs). Although HORMAD1 is known to be depleted from synapsed chromosome axes, this appears to be a gradual process, since with the sensitive dSTORM microscopy we still detected HORMAD1 localization on synapsed regions in early pachytene nuclei (Fig 1F and 1G) [52]. Similar to the HORMAD1 localization on unsynapsed chromosome axes, we also observed the double axes pattern along each of the lateral elements in synapsed regions (Fig 1H). A randomized control showed that, when localizations present in the measured indicated region were randomly repositioned in the same area, the double axes were not visible which suggests that the observed double axes pattern of HORMAD1 is a specific observation (Fig 1I).

We next investigated HORMAD1 localization in its 3D-environment using 3D-dSTORM. This analysis showed two HORMAD1 axes in both frontal and axial view, indicating the formation of two rod-like structures that run parallel in a zygotene nucleus (Fig 1J). Combined with the 2D-dSTORM data that include SYCP3 or SYCP1, we propose that two HORMAD1 rods run along each side of the axial elements, and also in the same plane with SYCP1 in lateral elements of synapsed SC (Fig 1K).

### Three-color dSTORM imaging of recombination foci in wild type and HORMAD1-deficient spermatocytes

The SC has a structural role in meiotic prophase, and also serves as a platform to facilitate meiotic DSB repair. We recently used super-resolution microscopy to analyze the accumulation

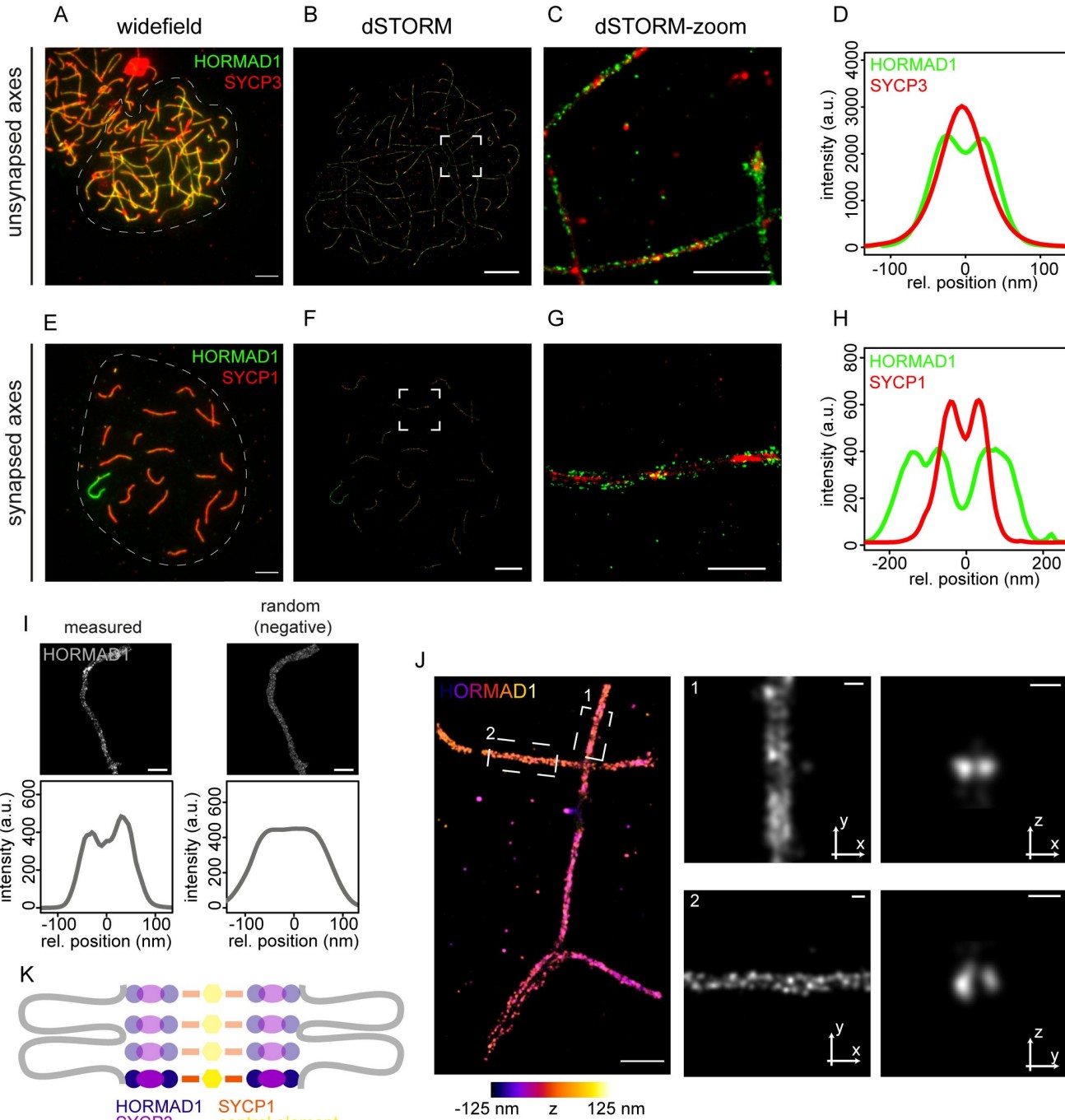

**Fig 1. Two-color dSTORM images of nuclear surface spread spermatocytes immunostained for synaptonemal complex proteins on synapsed and unsynapsed chromosomal axes.** A) Widefield image of a mouse zygotene nucleus immunostained for HORMAD1 (green) and SYCP3 (red). The indicated region (dashed line) is visualized with dSTORM in B. C) Close-up of boxed region shown in B. D) Average intensity-profiles of HORMAD1 (green) and SYCP3 (red) at unsynapsed chromosomal axes (from 3 zygotene cells, 8 chromosome regions). E) Widefield image of a mouse early pachytene nucleus immunostained for HORMAD1 (green) and SYCP1 (red). The indicated region (dashed line) is visualized with dSTORM in F). G) Close-up of boxed region shown in F. H) Average intensity-profiles of HORMAD1 (green) and SYCP1 (red) at synapsed chromosomal axes (from 2 early pachytene cells, 8 chromosome regions). I) Random control for two axes of HORMAD1. Localizations from the measured data were randomly positioned in the same area whereafter the intensity profiles were measured. J) 3D-dSTORM of HORMAD1 in zygotene nucleus. The z-position is specified with the colored scale bar. The panels on the right represent the area 1 (top panels) and 2 (lower panels) indicated on the left. For each area, the signal in the XY plane (middle) and the XZ/YZ plane (right) is enlarged. The two indicated stretches show a representative frontal view and a single slice of the axial view. K) Cartoon of HORMAD1 position in the SC. Other components of the SC are depicted in different colors. The chromatin is schematically depicted in grey. Scale bars represent 5 µm (A,B,E,F), 500 nm (C,G,J—left panel) and 100 nm (I and J—middle and right panels). All separate intensity profiles used to generate D and H are shown in S1 and S2 Figs.

patterns of the recombinases RAD51 and DMC1 during meiotic prophase [31]. In the current study, we asked how the recombinase patterns would be affected in the absence of HORMAD1, anticipating that this might help in explaining the functional significance of the identified recombinase accumulation patterns and dynamics during male meiotic prophase. In contrast to the two-color dSTORM analyses in our previous publication, we now performed three-color dSTORM imaging on wild type and *Hormad1*$^{-/-}$ spermatocytes immunostained for RAD51, DMC1 and an axis-component (SYCP3 or HORMAD1) (Fig 2A–2D). We also designed an automated method to select regions of interest (hereafter ROIs) in which RAD51-DMC1 foci are (partially) overlapping with SYCP3 or close to it (for details see Material and Methods, Figs 2E and S10). In wild type spermatocytes, the mean recorded number of ROIs per meiotic stage (128 foci in leptotene, 145 foci in early zygotene, 123 foci in mid zygotene, 93 foci in late zygotene and 58 foci in early pachytene) corresponded well with foci/DSB numbers reported in literature [53](Fig 2F). Since *Hormad1*$^{-/-}$ spermatocytes show synapsis defects, the substages were named leptotene-like, zygotene-like and early-pachytene-like based on short axes, long axes without synapsis and long axes with some synapsis, respectively. We measured an average of 51 foci in leptotene-like, 49 foci in zygotene-like and 31 foci in early pachytene-like in *Hormad1*$^{-/-}$ spermatocytes showing a reduction in number of foci compared to wild type as previously shown [34].

The dSTORM analyses resulted in the generation of two datasets for wild type (SYCP3-RAD51-DMC1 with 2717 ROIs from 23 nuclei of three animals, and HORMAD1-RAD51-DMC1 with 3245 ROIs from 32 nuclei of three animals, S1 Table) and one dataset for *Hormad1*$^{-/-}$ (SYCP3-RAD51-DMC1 with 1527 ROIs from 36 nuclei of three animals, S1 Table). For analyses that were independent of HORMAD1 or SYCP3, both wild type datasets were combined (5962 ROIs of 55 nuclei, S1 Table). Details of the ROIs can be found in S1 Table.

Within a ROI, multiple clustered localization events of each of the recombinases could be present in distinct subregions. To distinguish between foci in confocal images and these subregions of clustered localization events in dSTORM images, we termed these separate RAD51- and DMC1-positive subregions nanofoci, as introduced by Slotman et al. [31]. To compare the distribution of the defined nanofoci within nuclei of wild type and *Hormad1*$^{-/-}$ spermatocytes, we performed a nearest neighbor analysis using the center of all RAD51 and all DMC1 nanofoci separately (all stages combined). The average of the nuclei was visualized in a histogram showing nearest neighbor distances for RAD51 and DMC1 (Fig 2G–2N). This analysis revealed two preferred distances for RAD51 as well as DMC1: for wild type, a nearest neighbor distance between 150–450 nm was observed for 21.7% and 33.0% of the RAD51 and DMC1 nanofoci, respectively (Fig 2G and 2H), and a second preferred distance between 750–1050 nm was observed for 22.6% and 16.6% of the RAD51 and DMC1 nanofoci, respectively (Fig 2G and 2H). In the absence of HORMAD1, only 8.4% (RAD51) and 7.1% (DMC1) of the nanofoci showed a nearest neighbor distance frequency in the range between 750–1050 nm, while the frequency of the closer nearest neighbor distance peak was similar to wild type (Fig 2I and 2J). This finding can most likely be explained by the lower number of ROIs in *Hormad1*$^{-/-}$ spermatocytes, and to assess this we performed a nearest neighbor analysis on a subset of the wild type data to mimic the lower number of ROIs in the knockout. Since the nearest neighbor analysis was performed on the nanofoci coordinates, there were two ways to create a subset of the wild type data: ROI-based or nanofoci-based (Fig 2K–2O). In the ROI-based subset, we randomly selected a knockout-like number of ROIs from the wild type dataset and used the nanofoci coordinates within these ROIs for the nearest neighbor analysis. In the nanofoci-based subset, we randomly selected a knockout-like number of nanofoci from the wild type dataset and used these nanofoci coordinates for the nearest neighbor analysis. Thus,

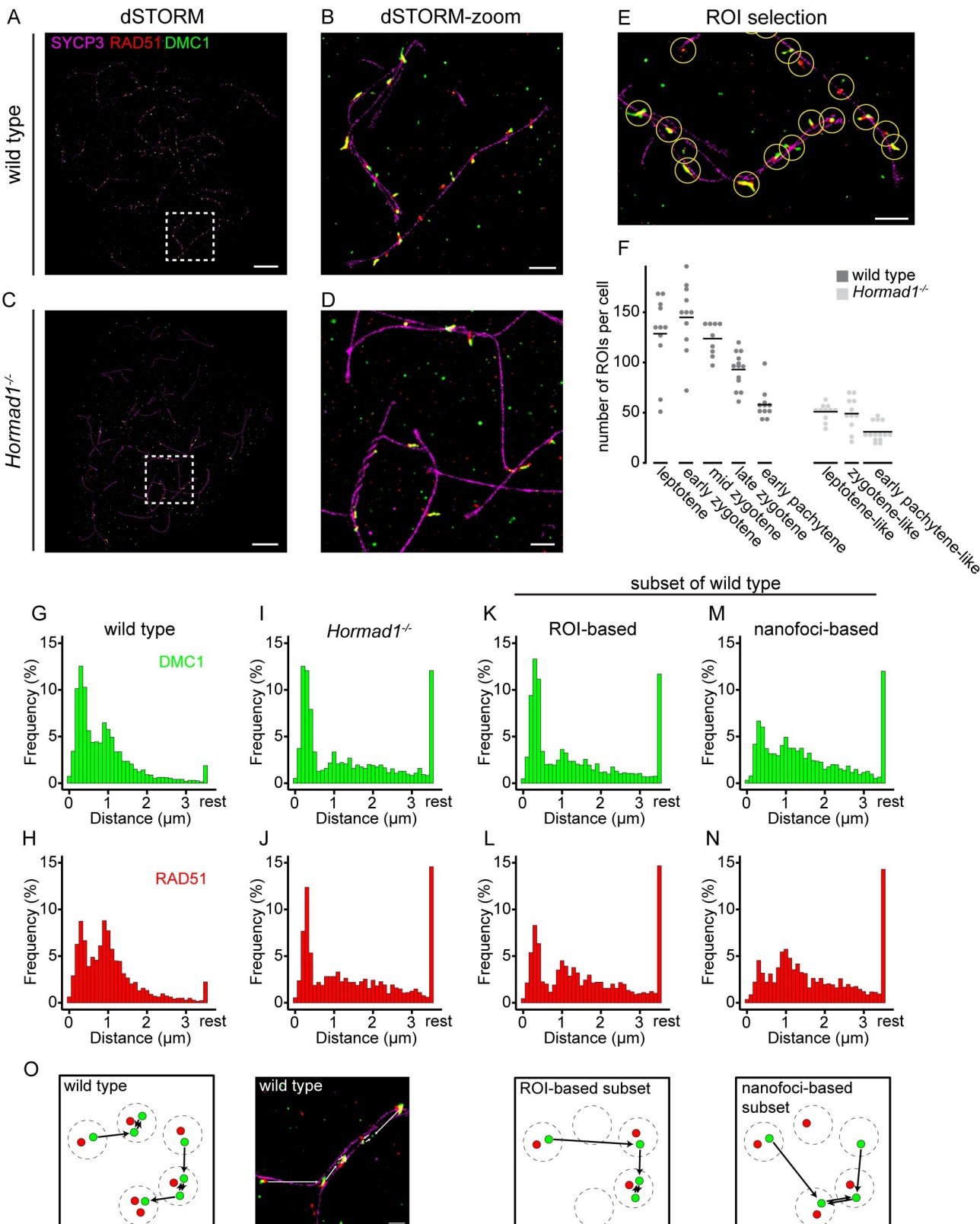

**Fig 2. Three-color dSTORM images of wild type and *Hormad1*⁻ᐟ⁻ nuclear surface spread spermatocytes immunostained for SYCP3-RAD51-DMC1.** A) dSTORM and B) close-up of boxed region shown in A) image of wild type mid-zygotene nucleus immunostained for

SYCP3 (magenta), RAD51 (red) and DMC1 (green). C) dSTORM and D) close-up of boxed region shown in C) image of *Hormad1*$^{-/-}$ early pachytene-like nucleus immunostained for SYCP3 (magenta), RAD51 (red) and DMC1 (green). E) Example of semi-automatic ROI selection. F) Dotplot of number of ROIs per nucleus at different meiotic stages for the wild type and *Hormad1*$^{-/-}$ dataset. Horizontal line represents the mean. G-N) Histogram of relative frequency distribution of nearest neighbor distances between nanofoci of wild type DMC1 (G) and RAD51 (H), *Hormad1*$^{-/-}$ DMC1 (I) and RAD51 (J), a ROI-based subset of wild type DMC1 (K) and RAD51 (L) and a nanofoci-based subset of wild type DMC1 (M) and RAD51 (N). DMC1 is indicated in green and RAD51 in red. Distances were binned in 100 nm bins, distances larger than 3.5 μm were labelled as rest. O) Cartoon and representative image of the nearest neighbor analysis with the two different wild type subsets. DMC1 nanofoci are depicted in green and RAD51 nanofoci in red. Dashed circles indicate a ROI and for every DMC1 nanofocus the shortest distance to the nearest DMC1 neighbor is indicated by a line with an arrow. Scale bars in A,C and B,D,E and O represent 5 μm, 1 μm, and 500 nm respectively. Numbers of nanofoci can be found in S2 Table.

in the nanofoci-based set, only one nanofocus or subset of nanofoci from a certain ROI may be selected. The ROI-based subset of wild type showed a similar distance distribution pattern as observed for *Hormad1*$^{-/-}$ (Fig 2K and 2L). This confirms that the decreased frequency observed for the nearest neighbor distance between 750–1050 nm in the knockout can be explained by the reduced number of ROIs, and that this distance represents the preferred distance between ROIs in wild type. The other preferred distance, between 150–450 nm, presumably corresponds to the preferred distance of nanofoci within a ROI. Indeed, the subset of the wild type data based on the nanofoci (where the ROI-structures were not taken into account) showed a loss of this short preferred distance (Fig 2M and 2N). A limitation of this analysis is that the position of ROIs along the synaptonemal complex was not taken into account, which could lead to a different distribution. Therefore, we performed an additional analysis using the manually drawn axes in zygotene and early pachytene nuclei. We defined coordinates along these lines by determining the intersection between a line drawn from the center of mass from each nanofocus to the nearest SC or axial element. The frequency distribution of nearest neighbor distances between these coordinates along the tracked axes showed a very similar distribution to that of the nearest neighbor analysis of nanofoci only, although a small peak around 750–1050 nm could still be discerned (S3 Fig). This may in part be due to the fact that more data points are lost in this type of analysis (axes that have only a single ROI associated with it are more frequent in the knockout, and are excluded here), but may also reveal some remaining organized distribution of DSBs in hotspot-rich regions. Together, with the previously observed preferred distance between foci observed using the confocal microscope (~800 nm, [31]), this strongly suggests that the signals observed in each ROI represent all RAD51 and DMC1 loading on ssDNA resulting from a single DSB at the moment of cell fixation.

## Precocious appearance of late RAD51 and DMC1 $D_xR_y$ configurations in absence of HORMAD1

Consistent with our previous research, we observed many different configurations of RAD51 and DMC1 (Fig 2A–2E and [31]). For further analysis, binary images with discrete RAD51 and DMC1 nanofoci were generated from the selected ROIs (Fig 3A, see Material and methods, [31]). We categorized the configurations ($D_xR_y$) based on the number of DMC1 nanofoci (x) and number of RAD51 nanofoci (y). In accordance with our previous data [31], D1R1 (69%) was the most frequent configuration, followed by D2R1 (15%) and D1R2 (6%) (Fig 3B). In the absence of HOR-MAD1, this frequency distribution was not significantly different (Fig 3B). As meiotic prophase progressed, the fraction of D1R1 decreased from 78% to 64%, while the D2R1 fraction increased from 10% to 17% (Fig 3C, p < 0.0001 and p = 0.0001 respectively (two-sided t-test)). The D1R2 fraction was stable over time. These dynamics correspond well to our previous observations with two-color dSTORM and different fluorophores [31]. In *Hormad1*$^{-/-}$ spermatocytes, we did not observe these dynamics as the cells progressed from the leptotene-like to the early pachytene-like stages (Fig 3D). When comparing wild type with knockout in leptotene, the knockout showed a

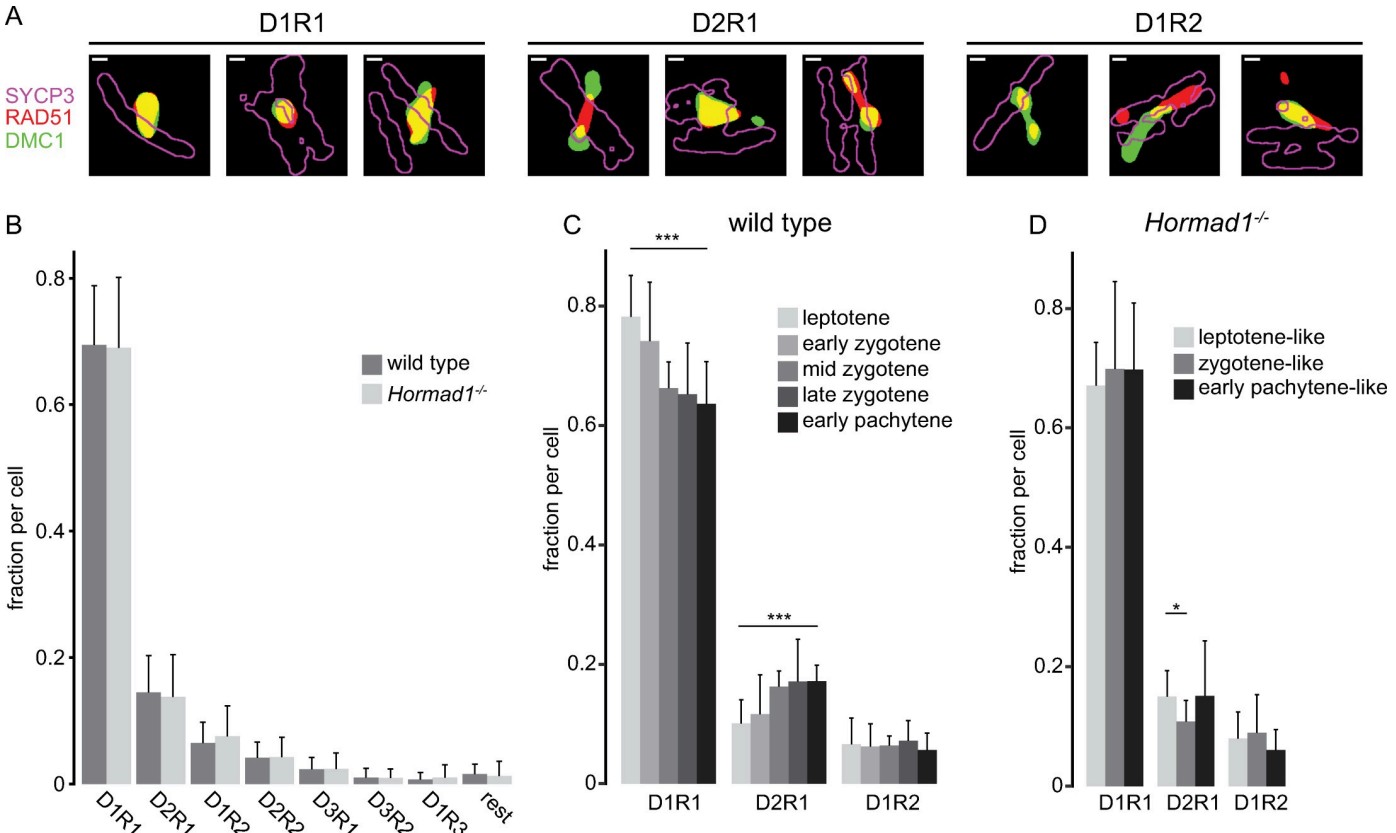

**Fig 3. RAD51 and DMC1 configuration in absence of HORMAD1.** A) Compilation of three binary representations of kernel density estimation configurations of single ROIs for each of the three main types (left panels D1R1, middle panels D2R1 and right panels D1R2). For SYCP3 only the outline is depicted in magenta, RAD51 is depicted in red and DMC1 in green. B) Barplot of fraction of ROIs with the indicated configuration of RAD51 and DMC1 nanofoci for wild type (n = 55 nuclei) and *Hormad1*<sup>-/-</sup> (n = 36 nuclei). Barplot of fraction of D1R1, D2R1 and D2R1 foci per nuclei per stage for wild type (C) and *Hormad1*<sup>-/-</sup> (D). Scale bars in A represent 100 nm. Error bars indicate standard deviation, p-values can be found in S2 Table.

lower frequency for D1R1 (p = 0.0015 (two-sided t-test)) and a higher frequency for D2R1 (p = 0.01 (two-sided t-test)), resulting in a more pachytene-like situation in the early leptotene-like stage of the knockout. This suggests that in wild type, D1R1 at least partially evolves into D2R1, and that the D1R1-D2R1 transition may occur prematurely in absence of HORMAD1.

It should be noted that we only selected ROIs if at least one nanofocus of each protein was present to reduce background. Still, true foci that contain only one of the recombinases likely also exist and are excluded from the analyses through this approach. To ensure that we would not miss significant information because of this choice, we also performed analyses without this (extra) condition in the automated ROI detection. From this, we observed that D0R1 and D1R0 represented a small fraction in all nuclei, and no changes in their frequency over time were observed (S4 Fig, S2 Table). The fraction of D0R1 and D1R0 was increased in absence of HORMAD1, which is consistent with the presence of fewer true foci (S4 Fig, S2 Table).

## Shape characteristics of D1R1, D2R1, and D1R2 configurations during meiotic prophase

Subsequently, we investigated the shape characteristics of the nanofoci present in D1R1, D2R1 and D1R2. Due to asymmetry observed in D2R1 and D1R2 organization, far and close nano-foci could be defined as follows: the closest nanofocus to the nanofocus of the other protein is

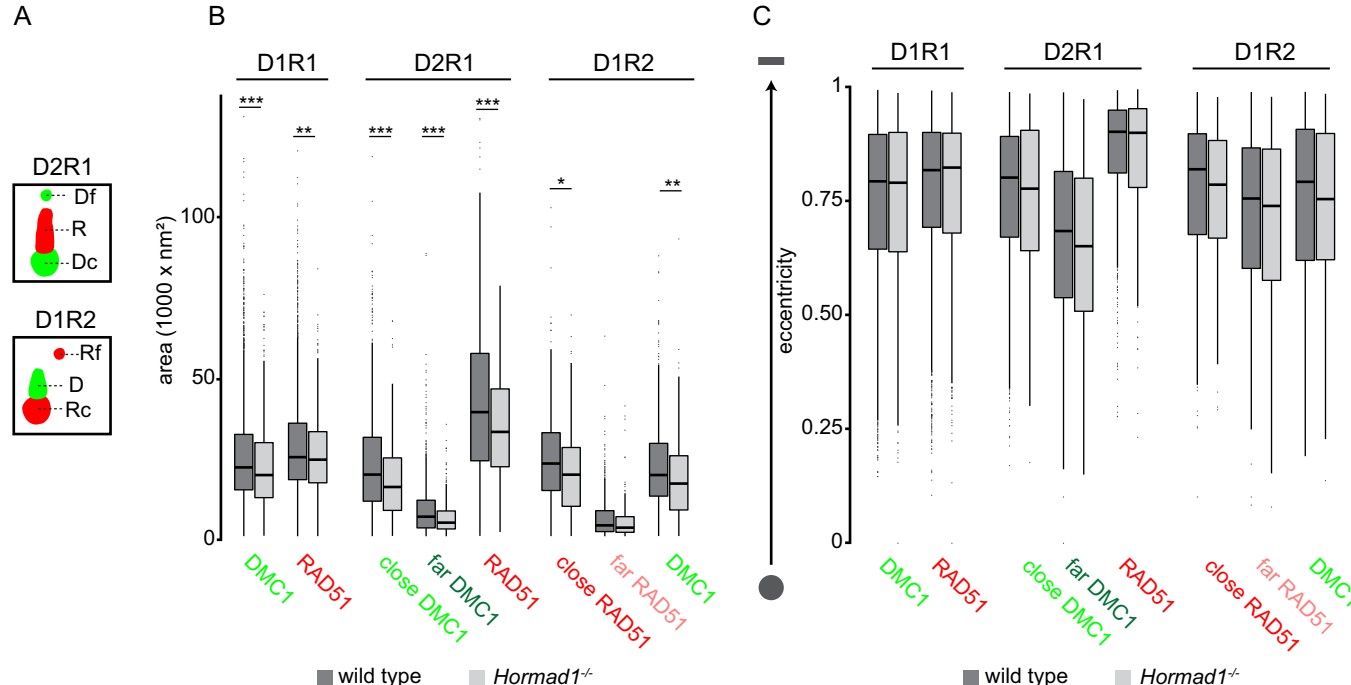

**Fig 4. Characterization of DSB configurations based on shape features of individual nanofoci.** A) Schematic representation of the nomenclature of each nanofocus within D2R1 and D1R2. RAD51 (R) and DMC1 (D) nanofoci within D2R1 and D1R1 are defined as follows: the closest nanofocus to the nanofocus of the other protein is called the close nanofocus (Dc, Rc) and the other nanofocus is called the far nanofocus (Rf, Df). B) Boxplot of area of RAD51 and DMC1 nanofoci for D1R1, D2R1 and D1R2 configurations for wild type and *Hormad1*[-/-]. C) Boxplot of eccentricity of RAD51 and DMC1 nanofoci for D1R1, D2R1 and D1R2 configurations for wild type and *Hormad1*[-/-]. For D2R1 and D1R2, a distinction was made between far and close nanofoci as defined in A. To visualize the relation between eccentricity and shape we depicted two schematic examples on the left: a low eccentricity value represents a round/sphere shape and a high eccentricity value represents a more oval/line shape.

called the close-nanofocus and the other nanofocus is called the far-nanofocus ([31], Fig 4A). To start with, the area of the nanofoci was measured and revealed two marked general features. First, the far nanofoci of both D2R1 and D1R2 were always much smaller than the other nanofoci within the ROI, which is in accordance with our previous observations [31]. Second, the RAD51 nanofocus of D2R1 was larger compared to the RAD51 nanofocus of D1R1 ROIs (Fig 4B). These relative differences in area size between nanofoci in ROIs were similar in wild type and *Hormad1*[-/-] spermatocytes, but overall, the RAD51 and DMC1 nanofocus area was found to be significantly smaller in the knockout compared to wild type in almost all nanofoci categories (Fig 4B, S2 Table). As meiotic prophase I progresses, the area of (close) RAD51 and DMC1 nanofoci of D1R1 and D2R1 showed small increases between leptotene and early pachytene (S5 Fig, S2 Table). For D1R2 no clear size changes over time were observed in wild type. The size increase over time for RAD51 in D1R1 was lacking in the absence of HORMAD1 (S5 Fig). Similarly, in the knockout the RAD51 nanofocus of D2R1 did not increase in size over time. Conversely, the far RAD51 in D1R2 was smaller in size in *Hormad1*[-/-] early pachytene-like spermatocytes compared to wild type early pachytene spermatocytes. To see if these differences in area within ROIs and between wild type and *Hormad1*[-/-] were connected to a difference in shape, the eccentricity was calculated. Indeed, the larger RAD51 nanofocus was more elongated in D2R1 compared to the somewhat smaller RAD51 nanofocus of D1R1 (Fig 4C). Moreover, the small far DMC1 of D2R1 showed the most round shape (indicated by the eccentricity value), suggesting a specific organization of nanofoci in D2R1. However, the overall smaller nanofocus size in *Hormad1*[-/-] spermatocytes was not associated with overall changes in eccentricity (Fig 4C). Additionally, for

both wild type and *Hormad1*$^{-/-}$ spermatocytes, RAD51 and DMC1 nanofoci of D1R1 became slightly more elongated as prophase progressed (S6 Fig). However, in D2R1 foci of HORMAD1--deficient spermatocytes, the close DMC1 was less eccentric in leptotene-like nuclei compared to later stages, and this was not observed for wild type. The smaller, far DMC1 nanofocus significantly decreased in eccentricity over time in *Hormad1*$^{-/-}$, and although statistical significance was not reached in wild type, the trend was the same.

The area and eccentricity describe the shape of $D_xR_y$ configurations, but information about the organization of nanofoci relative to each other within a configuration is lacking. To analyze this aspect, we studied if there were consensus patterns of D2R1 and D1R2 organization by rotating each ROI as described before [31]. In short, close DMC1 or close RAD51 was used as center (anchor) for D2R1 and D1R2 rotations respectively, and the corresponding single RAD51 and DMC1 nanofoci were rotated until they aligned above the anchor. Subsequently, consensus localization patterns of the far-clusters were determined by summation of all foci within the indicated configuration. The summed kernel density estimation image showed that the far DMC1 nanofocus of D2R1 preferred a location above RAD51 (Fig 5A), suggesting a dumbbell structure whereby the two DMC1 nanofoci are connected via the RAD51 nanofocus. This was in contrast to D1R2, where the far RAD51 nanofocus was more spread around the center, although a preferred dumbbell-structure is still present (Fig 5A). It has to be mentioned that, when displaying a heatmap-density plot of the localizations, all localizations of the indicated configurations are combined. This could lead to a biased representation of nanofoci with many localization events. Therefore, a representation that is independent of the number of localization events for a single nanofocus is shown as center-of-mass dot-plot of the far-cluster (Fig 5A). This still reveals the dumbbell-structure for D2R1 but no longer for D1R2, which indicates that the D1R2 configurations which do show a dumbbell-structure contained a relatively high density of localizations within their far-nanofocus compared to the nanofoci with a different orientation relative to close RAD51. These consensus patterns obtained by rotation analysis were not different in absence of HORMAD1 as compared to wild type (Fig 5B).

To investigate if aspects of these consensus patterns changed over time, we measured the distance between nanofoci within D1R1, D2R1 and D1R2. For D1R1, we observed that the distance between RAD51 and DMC1 increased from 39.7 nm to 44.7 nm (median, p = 0.01 (Wilcoxon signed-rank test)) as meiotic prophase progresses in wild type. In contrast, this distance decreased from 54.2 nm to 42.8 nm in absence of HORMAD1 (Fig 5C, median, p <0.001 (Wilcoxon signed-rank test)). For D2R1 three distances were measured, of which the distance from far DMC1 to RAD51 and that of close DMC1 to far DMC1 showed a significant increase over time (Fig 5E and 5F, p = 0.045 and p = 0.024 respectively (two-sided t-test)). In *Hormad1*$^{-/-}$ spermatocytes these structural changes for D2R1 were not observed (Fig 5D–5F). Conversely, for D1R2, stage-dependent changes of distances between nanofoci were observed for *Hormad1*$^{-/-}$ but not for wild type (Fig 5G–5I). The distances from close RAD51 to DMC1 (Fig 5G, p = 0.0086 (Wilcoxon signed-rank test)) and close RAD51 to far RAD51 (Fig 5I, p = 0.047 (two-sided t-test)) significantly decreased over time in *Hormad1*$^{-/-}$ D1R2, similar to the observed decreasing distances between DMC1 and RAD51 in D1R1 foci of the knockout.

Taken together, the overall structure of the D1R1, D2R1 and D1R2 foci is maintained in the knockout, but all nanofoci are slightly smaller and there are some further more subtle changes as schematically summarized in Fig 6.

## RAD51 and DMC1 configurations in relation to the synaptonemal complex

Besides the spatial organization of RAD51 and DMC1 nanofoci within one DSB focus, the three-color dSTORM imaging also revealed detailed information about the relation of DSB

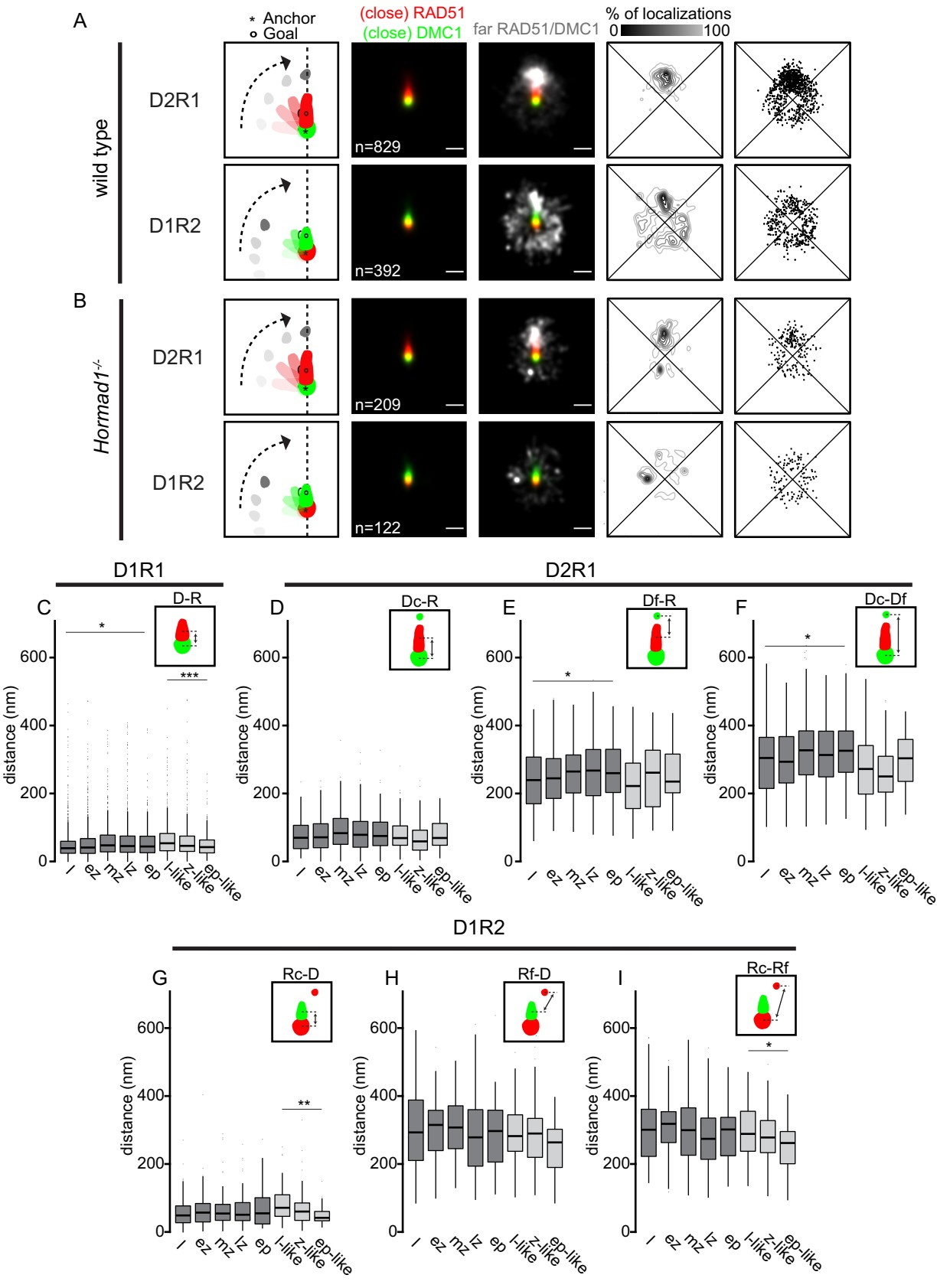

**Fig 5. Consensus pattern of D1R1, D2R1 and D1R2 configurations for wild type and *Hormad1*<sup>-/-</sup>.** Rotation analysis of all D2R1 and D1R2 foci of wild type (A) and *Hormad1*<sup>-/-</sup> (B). From left to right: Images were rotated as indicated in the schematic drawing whereby the anchor (*) indicates the center and the goal (o) was rotated until it was vertically aligned with the center. Summed kernel density estimation images of all D2R1 or D1R2 foci with (close) RAD51 (red) and (close) DMC1 (green). The same image with an additional channel representing far RAD51 or far DMC1 (white). Heatmap-style density plot of far nanofocus localizations. Dot-plot representing the center of mass of each far nanofocus. Scale bars represent 250 nm. Boxplot of distances between the indicated nanofoci per stage within configurations D1R1 (C), D2R1 (D-F) and D1R2 (G-I) in wild type and *Hormad1*<sup>-/-</sup> spermatocytes. Leptotene (l), early zygotene (ez), mid zygotene (mz), late zygotene (lz), early pachytene (ep), leptotene-like (l-like), zygotene-like (z-like), early pachytene-like (ep-like). p-values can be found in S2 Table. n in A) and B) indicates the number of ROIs.

foci to the axes. Therefore, we investigated the position of axes in relation to RAD51 and DMC1. Similar to the rotation analysis for the far-nanofoci, we also performed a rotation analysis to obtain a consensus pattern of SYCP3 in relation to RAD51 and DMC1 for D1R1, D2R1 and D1R2 (Fig 7A). In wild type D1R1, SYCP3 was localized closer to the rotated RAD51 center compared to the DMC1 center, indicating that SYCP3 is closer to RAD51 than DMC1. This also applied to D2R1, where SYCP3 was even more clearly located closer to the RAD51 center compared to the close DMC1 center, suggesting that the dumbbell structure is oriented so that RAD51 is centered on SYCP3 and the two DMC1 nanofoci are positioned more distal from the SC-axis center. For D1R2, in which close RAD51 was used as center and DMC1 as

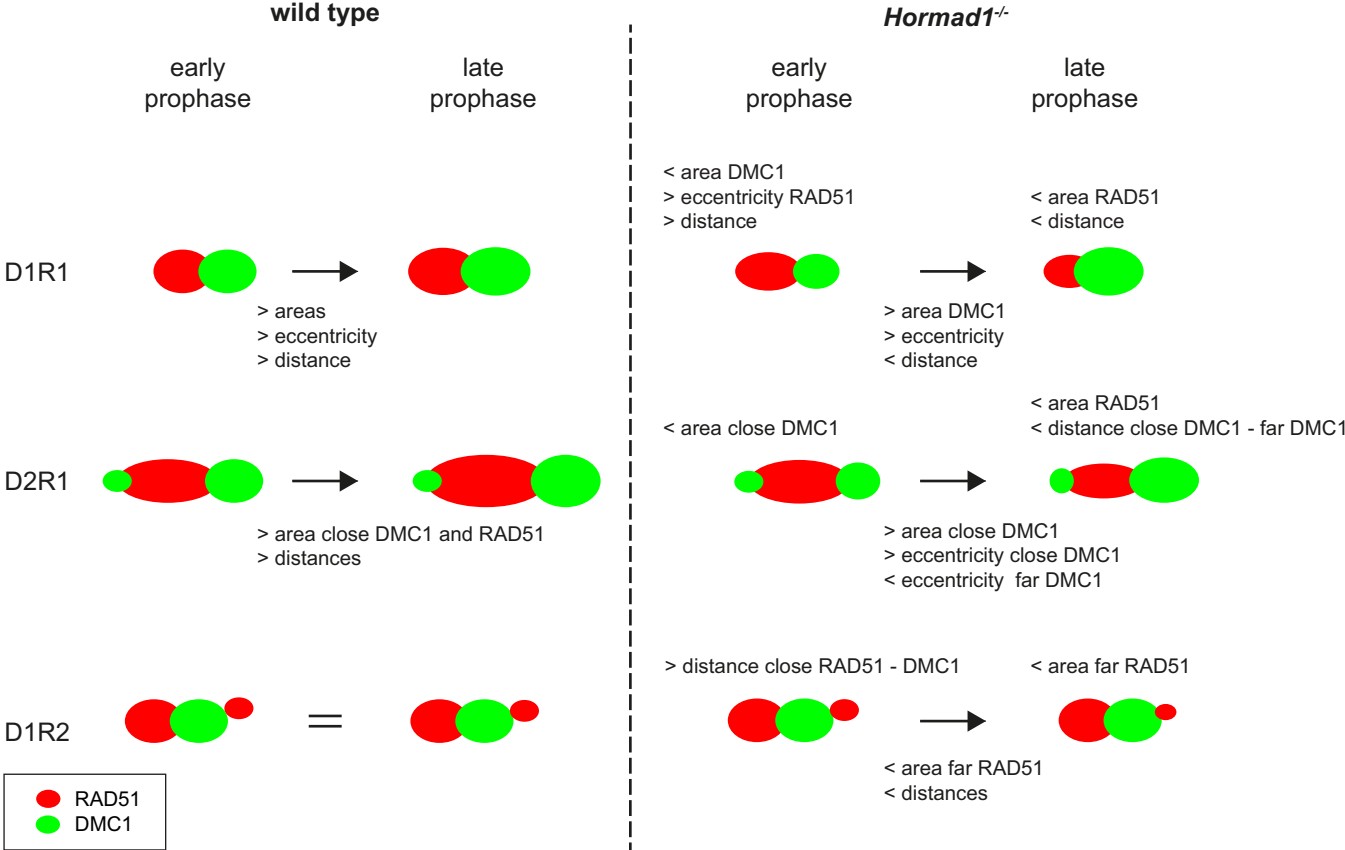

**Fig 6. Schematic overview of alterations in D1R1, D2R1 and D1R2 configurations during meiotic prophase for wild type and *Hormad1*<sup>-/-</sup>.** Summary of alterations in area, eccentricity and internal distances. The left images schematically represent the consensus configurations in wild type, and how they change over time indicated by the text below the arrows. For D1R2 no change over time was observed, indicated by =. On the right side, the same is shown for *Hormad1*<sup>-/-</sup>, and differences with wild type that exist in early or late prophase are described above the relevant schematic configurations. RAD51 is red and DMC1 is green.

goal, SYCP3 was located closer to RAD51, again agreeing with the observation that RAD51 is closer to the SC compared to DMC1. In *Hormad1*$^{-/-}$ spermatocytes, this rotation analysis revealed the same results as described for wild type (Fig 7B). Additionally, this overall consensus pattern was not changed when HORMAD1 was used as axis-marker (S7A Fig).

To confirm this consensus pattern of the SC and recombinases, we used an additional approach to identify the location of the recombinases in relation to the SC. We performed an adaptation of the distance analysis as described by Slotman et al. (2020) whereby in this case we used the SYCP3 signal from the dSTORM-image instead of SIM data. Briefly, we manually drew a line through the center of SYCP3 (e.g. for an unsynapsed region one line through the center of the axial element and for a synapsed region two lines through the center of each lateral element). These lines were used to calculate the minimal distance to each RAD51/DMC1 nanofocus (center of mass) to SYCP3 (Fig 7C). Distances larger than 500 nm were excluded from this analysis. Taking all nanofoci together revealed a median distance of RAD51 and DMC1 to SYCP3 of 89.8 nm and 107.2 nm respectively, confirming the observation that RAD51 is closer to SYCP3 than DMC1 (Fig 7D, p < 0.001 (Wilcoxon signed-rank test)) [31,32]. In the absence of HORMAD1, RAD51 was also closer to the SYCP3-axis than DMC1 (median of 111.3 nm and 126.1 nm respectively, p = 0.005 (Wilcoxon signed-rank test)). As expected, overall, RAD51 nanofoci displayed a larger percentage of signal overlap with SYCP3 compared to DMC1, confirming the above described finding of RAD51 being closer to the axis (compare S7B Fig with Fig 7D). However, both RAD51 and DMC1 individually showed larger distances to the axis in the absence of HORMAD1 as compared to wild type (both p < 0.001 (Wilcoxon signed-rank test)). If we separate the data based on the stages of meiotic prophase, RAD51/DMC1 nanofoci in wild type localize closer to SYCP3 as cells transit from mid-zygotene to late zygotene. This may reflect closer localization of nanofoci to (one of) the lateral elements, compared to their association with axial elements (Fig 7E). Indeed, in the absence of HORMAD1, where there is little synapsis, this transition is not observed, and RAD51 and DMC1 distances did not change over time and were similar to those observed in the early stages of the wild type. This apparent movement closer to the axes over time is confirmed when a distinction is made between unsynapsed and synapsed regions (Fig 7F). Taken together, the difference between wild type and *Hormad1*$^{-/-}$ in distance of recombinase nanofoci to SYCP3 can be explained by the difference in degree of synapsis.

Next, we discriminated between the different nanofoci configurations ($D_xR_y$) to assess the distances to the axes in more detail. In accordance with previous research, there is no difference in relative frequency of recombinase configuration between synapsed and unsynapsed axes (S7C and S7D Fig). Since the synapsed dataset of the mutant becomes too small when the data are separated in this manner, we compared the distances of $D_xR_y$ nanofoci and chromosomal axes only for unsynapsed situations between the genotypes, and distances were compared between synapsed and unsynapsed regions only in wild type (Fig 7G). Comparing the distances between genotypes in unsynapsed situations showed no marked difference between wild type and knockout (only the DMC1 of D1R1 showed a slightly shorter distance to SYCP3 in *Hormad1*$^{-/-}$ compared to wild type at unsynapsed axes)(p = 0.0498 (Wilcoxon signed-rank test)). Almost all nanofoci localized closer to the axes in wild type, upon synapsis, and, remarkably, the shortest distance to the axes was observed for the RAD51 nanofocus in the D2R1 configuration (Fig 7G). We wondered whether this would be mainly represented by nanofoci localizing between synapsed axes, or on the outside, close to only one of the axes. To assess this, we determined the position of the nanofoci relative to each of the two lateral elements and scored whether the center of mass was inside or outside of the SC. This analysis showed that overall, the center of mass of the nanofoci was positioned more frequently outside the lateral elements than in between. Of all the nanofoci, RAD51 of D2R1 is most frequently

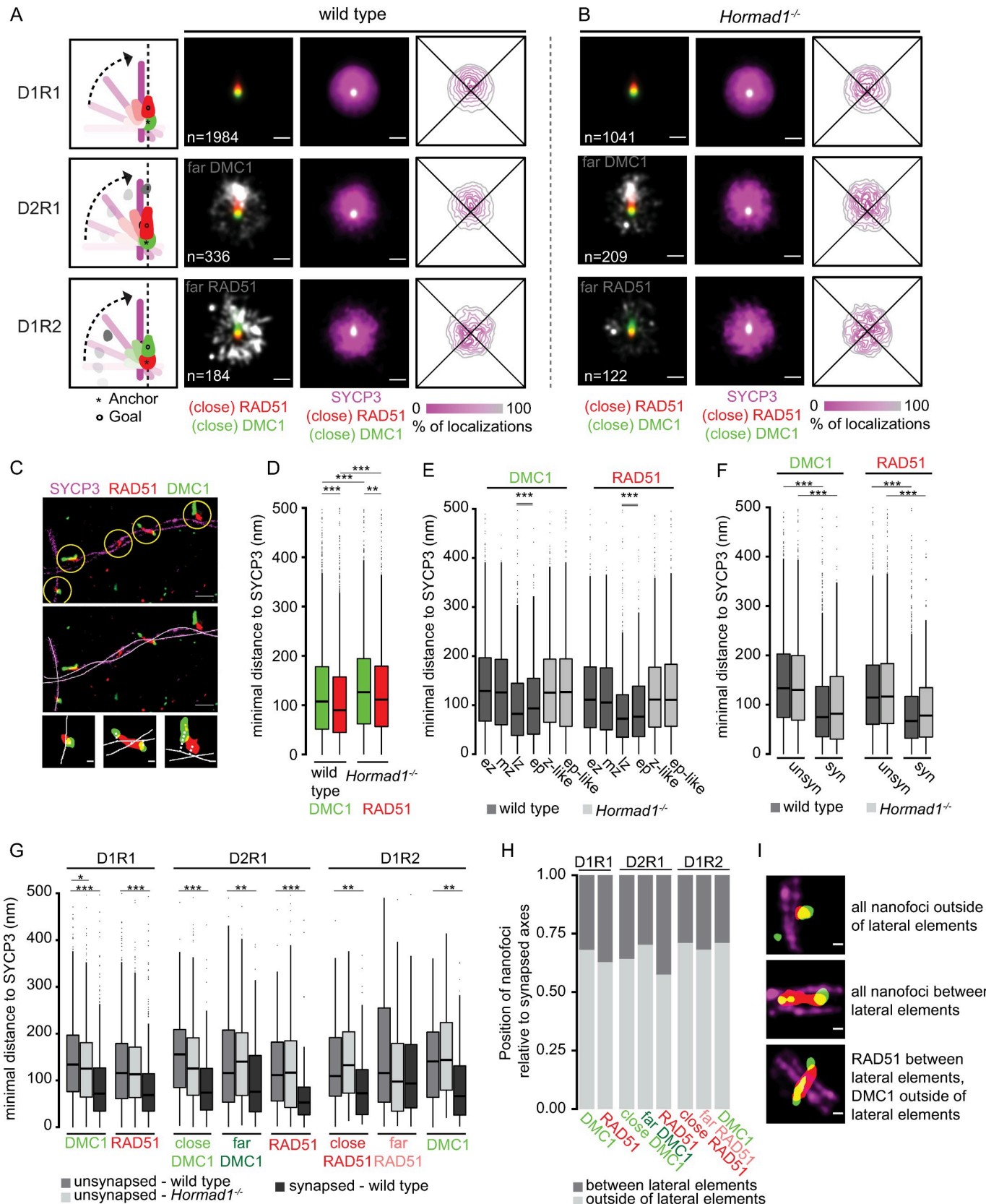

**Fig 7. Consensus pattern of synaptonemal complex and distances to SYCP3 for D1R1, D2R1 and D1R2 configurations for wild type and *Hormad1*<sup>-/-</sup>.**

Rotation analysis of all D1R1, D2R1 and D1R2 foci relative to the SYCP3 channel of wild type (A) and *Hormad1*<sup>-/-</sup> (B). From left to right: Images were rotated as indicated in the schematic drawing whereby the anchor (*) indicates the center and the goal (o) was rotated until it was aligned with the center. Summed kernel density estimation image of all D1R1, D2R1 or D1R2 foci with (close) RAD51 (red) and (close) DMC1 (green) combined with far-nanofocus (white) or SYCP3 (magenta). Heatmap-style density plot of SYCP3 localizations. C) Three-color-dSTORM image with SYCP3 (magenta), RAD51 (red) and DMC1 (green) with ROIs (yellow circle–top panel) and manually-drawn axis (white lines–middle panel). Three of the ROIs shown in the dSTORM-image are visualized as binary image with the manually-drawn axis. The minimal distance from the center of mass of the nanofocus to the manually-drawn axis is schematic visualized with the grey dashed line. D) Boxplot of minimal distance to SYCP3 axis of all DMC1 and RAD51 nanofoci for wild type and *Hormad1*<sup>-/-</sup>. E) Boxplot of minimal distance to SYCP3 axis of all DMC1 and RAD51 nanofoci per stage for wild type and *Hormad1*<sup>-/-</sup>. Early zygotene (ez), mid zygotene (mz), late zygotene (lz), early pachytene (ep), zygotene-like (z-like), early pachytene-like (ep-like). F) Boxplot of minimal distance to SYCP3 axes for RAD51 and DMC1 at unsynapsed and synapsed regions for both wild type and *Hormad1*<sup>-/-</sup>. Unsynapsed (unsyn) and synapsed (syn). G) Boxplot of minimal distance to SYCP3 axis of nanofoci of D1R1, D2R1 and D1R2 for wild type and *Hormad1*<sup>-/-</sup>. H) Barplot of position of nanofoci of D1R1, D2R1 and D1R2 for wild type relative to the lateral elements. I) Example images of the position of nanofoci relative to the lateral elements. SYCP3 (magenta), RAD51 (red) and DMC1 (green). Scale bars represent 250 nm (A, B), 500 nm (C—large panels) and 100 nm (C—small panels, I). Distances larger than 500 nm were excluded for the distance analyses. p-values can be found in S2 Table. Double lines below * in E) indicate that those two stages are significantly different from the other stages, but not from each other. n in A) and B) indicates the number of ROIs.

positioned between the two lateral elements (Fig 7H), which is in accordance with the observed distances to the SC and the results of the rotation analyses (Fig 7A). Thus, these data again argue for a D2R1 dumbbell structure with RAD51 located nearest to the center of the SC and the two DMC1 nanofoci located on each side, further away from the center. To perform a more specific analyses of orientation of D2R1 ROIs relative to the axes, we calculated the angle between a line fitted through the centers of mass of the three recombinase nanofoci within the ROI and a line fitted through the mask of SYCP3 signal (S7E Fig). The median was 48.2˚, and 42.2˚ in wild type and *Hormad1*<sup>-/-</sup>, respectively, but this change was not significant (S7F Fig, S2 Table). This indicates that indeed, the dumbbell structure often crosses the SC, but not precisely perpendicular.

## RAD51 and DMC1 configurations in specialized regions

Currently, it is still unknown how the cell selects DSBs to resolve into a crossover (CO) or non-crossover, and to what extent repair via the sister chromatid may occur. It is also uncertain whether these molecular outcomes would be somehow reflected in different RAD51 and DMC1 nanostructures. One certainty is that recombination is regulated such that each chromosome pair will develop at least one CO (reviewed by [54]). Since the X and Y chromosomes of males share homology only in the small pseudo-autosomal region (PAR), this requires special regulatory adaptations to secure formation of a crossover in this area [42,44,45,55,56]. In contrast, DSBs on the heterologous regions of the X or Y chromosome only have the sister-chromatid available for repair. Because of these special characteristics, we investigated the repair foci on the XY pair in more detail in all wild type late zygotene and early pachytene nuclei in which they could be unequivocally identified based on morphology (n = 24 nuclei including 220 sex chromosome ROIs and 1625 autosome ROIs). We observed that the fraction of D1R1 was lower along the X chromosome than on autosomes. In contrast, the fraction D2R1 was higher (Fig 8, both p = 0.03 (Fisher-exact test)). The decrease in D1R1 fraction was even more evident for ROIs along the Y chromosome (p = 0.01 (Fisher-exact test)), and this was associated with enrichment of more complex configurations (D2R2 and D1R3) compared to the autosomes. Surprisingly, in the PAR, the fraction of D2R1 was even more enriched than along the X, representing approximately 40% of all PAR ROIs (Fig 8, p = 0.02 (Fisher-exact test)). Note that these regions (Y chromosome and PAR) are relatively short in length compared to X chromosome and autosomes and therefore the number of foci that could be analyzed was relatively small. Still, these results suggest that the region where the obligate XY CO needs to occur more often displays D2R1 configurations.

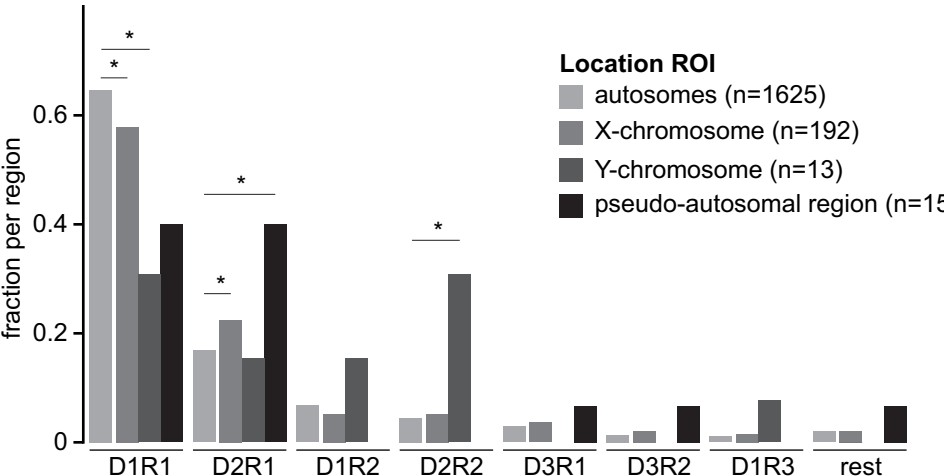

**Fig 8. Recombinase configurations at sex-chromosomes.** Barplot of fraction of RAD51 and DMC1 configurations of ROIs present on the X chromosome, Y chromosome, PAR and the autosomes of late-zygotene and early pachytene nuclei. p-values can be found in S2 Table. n indicates the number of ROIs.

## Increased coiling capacity of synaptonemal complex in the absence of HORMAD1

During the dSTORM experiments, we surprisingly observed that in absence of HORMAD1, the SC showed more coils (Fig 9A–9D). It has been known already for decades that the lateral elements of the SC show cross connections, and that this is caused by the fact that the SC has a helical structure [57]. The mean distance between coils for wild type and *Hormad1*$^{-/-}$ SCs was 2.1 μm and 1.0 μm, respectively (Fig 9E, p < 0.0001 (two-sided t-test)). In wild type, a chromosome axis has 2.71 coils on average, and the number increases with increasing chromosome length (S8A and S8B Fig).

To investigate if this phenotype is specific to HORMAD1, and related to synapsis and/or DSB formation, other synaptonemal complex mutants were investigated (*Sycp1*$^{-/-}$, *Hormad2*$^{-/-}$) as well as a mutant that lacked all meiotic DSBs (*Spo11*$^{-/-}$). For these analyses we used stimulated emission depletion (STED) microscopy (Fig 9F and 9G). Quantification of coiling was performed by measuring the length of synapsed chromosomal axes (or clear alignment of the chromosomal axes in case of *Sycp1*$^{-/-}$) and the number of coils (see Material and Methods), resulting in an average distance between coils per nucleus as parameter to assess increases or decreases in coiling. The results revealed that only *Hormad1*$^{-/-}$ SC showed a strong increase in coiling. However, in *Sycp1*$^{-/-}$ spermatocytes, in which the formation of transverse filaments is defective, the coiling frequency was less compared to wild type, and SPO11-deficient spermatocytes displayed a small increase in coiling (p = 0.026 (two-sided t-test)). *Hormad2*$^{-/-}$ spermatocytes showed a very slight but significant increased distance of 2.56 μm between coils (p = 0.03 (two-sided t-test)).

Using dSTORM, another SC-mutant, *Sycp3*$^{-/-}$, was investigated by staining for the N-terminus of SYCP1 (S8C–S8F Fig). Average line-profiles of wild type spermatocytes showed double axes for SYCP1, but in absence of SYCP3, separation of SYCP1 was less clear which precluded the analysis of the coiling-phenotype in this mutant with the current fluorescence microscopy tools (S8G Fig). Together, this suggests that SC coiling is inhibited by HORMAD1 and SPO11, and promoted by SYCP1.

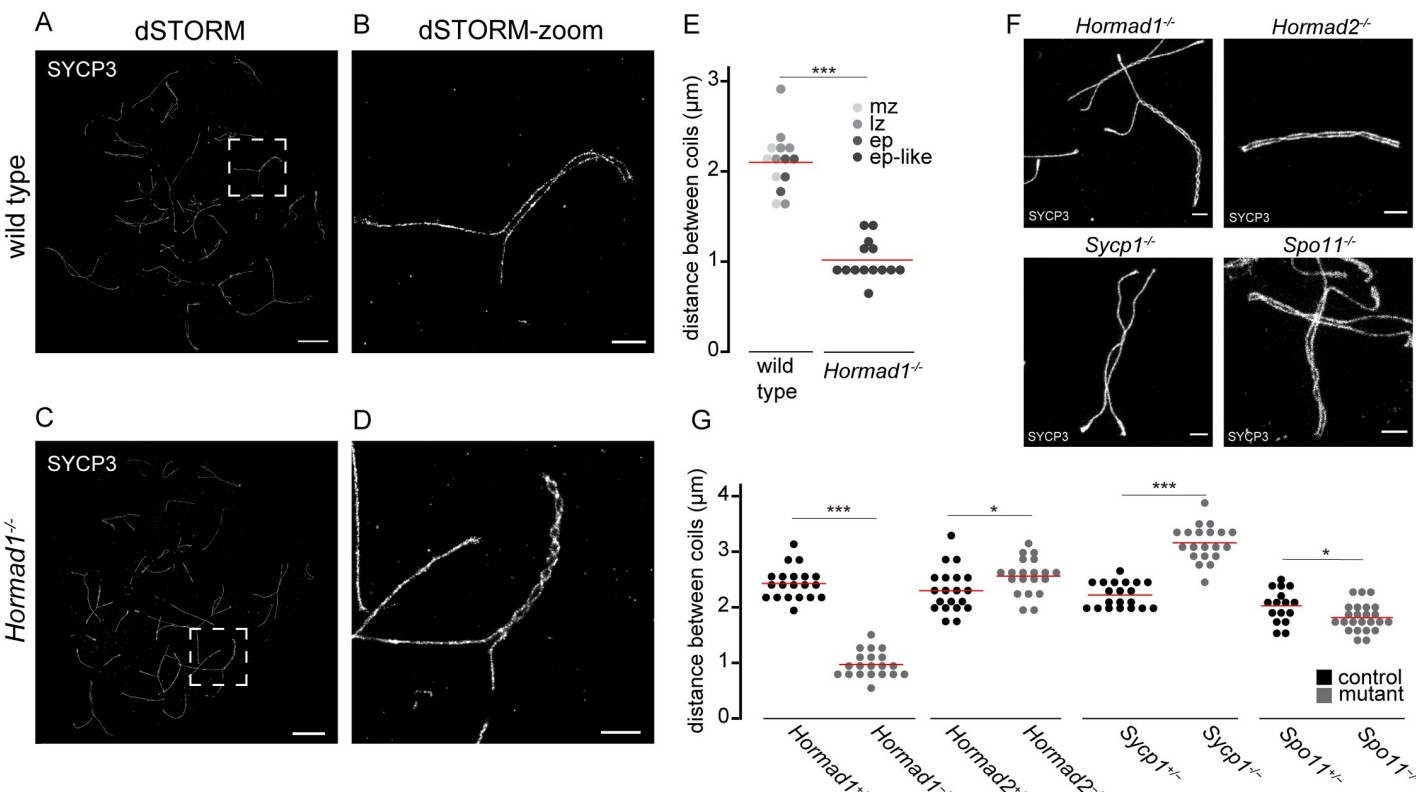

**Fig 9. Synaptonemal complex coiling on meiotic spread spermatocytes of several mutants investigated with dSTORM and STED microscopy.** A) dSTORM image of wild type mid-zygotene nucleus immunostained for SYCP3. B) Close-up of boxed region shown in A. C) dSTORM image of *Hormad1⁻/⁻* early pachytene-like nucleus immunostained for SYCP3. D) Close-up of boxed region shown in C. E) Dotplot of distance between coils of SYCP3 of wild type and *Hormad1⁻/⁻* in mid zygotene-early pachytene nuclei of wild type and early pachytene-like nuclei of *Hormad1⁻/⁻*. Color code represents the different meiotic stages. Mid zygotene (mz), late zygotene (lz), early pachytene (ep), early pachytene-like (ep-like). F) Representative STED images of meiotic spreads of *Hormad1⁻/⁻*, *Hormad2⁻/⁻*, *Sycp1⁻/⁻* and *Spo11⁻/⁻* late prophase nuclei immunostained for SYCP3. G) Dotplot of distance between coils for the mutants showed in F whereby each dot represents an average per nuclei. In both E and G, the red line indicates the mean. Scale bars in A,C and B,D,F represent 5 μm and 1 μm respectively. p-values can be found in S2 Table.

## Discussion

The purpose of this study was to use two- and three-color super-resolution dSTORM microscopy and the *Hormad1⁻/⁻* mouse model to obtain more insight in the functional significance of the RAD51-DMC1 accumulation patterns that can be observed with super-resolution microscopy. We observed that the previously described frequency distribution of $D_xR_y$ configurations was robustly reproduced using this different approach, involving three-color dSTORM and automated ROI-detection. In addition, we identified new features of the nanofoci and found that although these were only subtly changed in the absence of HORMAD1, some clear functional implications arise, as detailed below. Additionally, our analyses also revealed a new role for HORMAD1 in regulating the coiling of the synaptonemal complex (SC).

### Model for recombination intermediates during meiotic prophase

Our previous analyses suggested that a $D_xR_y$ configuration might represent one end of a meiotic DSB in many occasions, where the other end would be ~800 nm away. This was based on nearest neighbor analyses of foci distances in confocal images, combined with the super-resolution data from within the foci [31]. Here, we analyzed nanofoci distances and identified two preferred distances of ~300 nm (within a ROI) and ~900 nm (between ROIs). If both preferred

distances would generally represent distances between two ends, or between two other entities associated with a single DSB, we would expect that when less DSBs form, these two preferred distances would still be detected in similar relative frequencies. Using the *Hormad1$^{-/-}$* model to test this hypothesis, we almost completely lost the ~900 nm preferred distance in absence of HORMAD1. This observation does not support the hypothesis that the two ends of a DSB would often be at 800 (confocal) - 900 nm (dSTORM) distance, but rather seems to be the result of the lower number of DSBs in the mutant, resulting in larger average distances, and more variation in this distance. Therefore, we assume that each $D_xR_y$ configuration represents one DSB (Fig 10A), and the preferred distance of ~300 nm corresponds to the preferred RAD51-RAD51 and DMC1-DMC1 distances in the D1R2 and D2R1 configurations, respectively.

Interpretation of the $D_xR_y$ configuration in terms of loading patterns on DSB repair intermediates is still speculative. Our single repair focus analysis using super-resolution microscopy showed a broad variety of recombinase configurations, while ChIP-seq data of RAD51 and DMC1 in mouse spermatocytes showed a rather uniform pattern where DMC1 is located closer to the break site while RAD51 is closer to the dsDNA [32]. Here, we try to integrate findings from both approaches in a model for recombinase organization during the meiotic prophase (Fig 10B).

To start with the organization of RAD51 and DMC1 on one ssDNA end, we favor the hypothesis that DMC1 is located at the end of the ssDNA, while RAD51 is present more towards the dsDNA as suggested by Hinch et al. [32]. This is in line with previous findings [31,32], and with our observation that RAD51 is closer to the SC than DMC1. This is also consistent with the idea that DMC1 is involved in homology search while RAD51 provides a more structural role, staying close to the SC [58–60]. In addition, our results also showed that, upon synapsis, both recombinases move closer to the lateral elements, and that this relocation occurs independent of HORMAD1.

In all stages of meiotic prophase, D1R1 is the most frequent configuration. Since the DSB repair intermediates are expected to change as prophase progresses, we infer that the D1R1 configuration represents loading patterns on early as well as late repair intermediates. For the early D1R1 configuration we favor two possibilities, that may co-exist. The first possibility assumes that the recombinases are loaded on one end of the DSB and that the other end is occupied by other proteins (Fig 10B[i]). This configuration may emerge if recombinases rarely replace ssDNA binding proteins (RPA, MEIOB, SPATA22) simultaneously on each of the two ssDNA ends. This hypothesis would predict that a nanofocus of (at least one of) the ssDNA binding proteins would be observed together with the D1R1 configuration within a single ROI, in a three-color dSTORM analyses with matching antibodies. The second possibility assumes that both ends accumulate both RAD51 and DMC1, are organized in a parallel fashion due to preferred homotypic low-affinity interactions, and cannot be resolved with super-resolution microscopy (Fig 10B[ii]). This idea is more difficult to reconcile with the previously proposed idea that one end of the DSB might be more engaged in homology search compared to the other, more quiescent end [61–63].

As meiotic prophase progresses, the D2R1 frequency increases while the D1R1 frequency decreases, suggesting that a D1R1 can transform into a D2R1. In our model we propose this transition (Fig 10B). In addition, with the use of an improved ROI selection, an optimized imaging method and more analyzed ROIs, we observed a stronger dumbbell consensus structure for D2R1. It could be that the D2R1 configuration represents an organization of the DMC1 loaded regions of the ssDNA ends that are searching for homology, while the RAD51 filaments on the two ends are so close together -and relatively more often localized in between paired axes (~40%)—that they cannot be resolved (Fig 10B[iii]). However, the fact that the D2R1

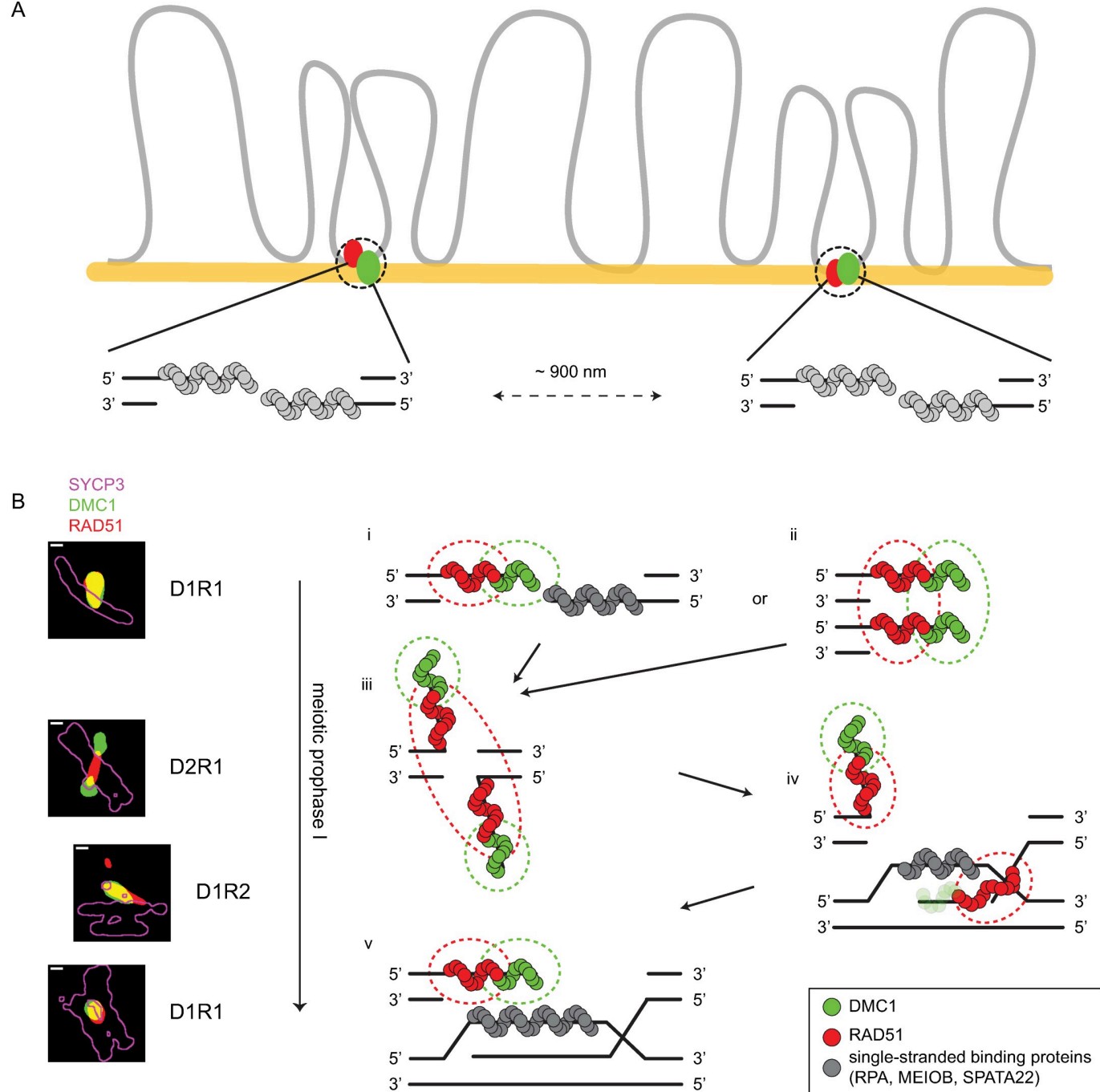

**Fig 10. Schematic model for recombinase loading at early steps of homologous recombination pathway in mouse meiosis.** A) Based on the nearest neighbor analysis of wild type and *Hormad1*[-/-] recombinase nanofoci, we propose that in fixed mouse spread nuclei of wild types, two DSBs are on average ~900 nm apart. This schematic drawing provides an impression of how the SC (yellow) and chromatin (grey) relate to the actual DSBs. B) The ssDNA ends of a DSB are first loaded by ssDNA binding proteins (grey). These proteins will be replaced by recombinases. This schematic model for the recombinase loading during meiotic HR relates super-resolution microscopy configurations with loading patterns of RAD51 (red) and DMC1 (green). Dashed lines indicate the signal of recombinases in immunofluorescent dSTORM microscopy. D1R1 is the most frequent configuration, followed by a D2R1 and D1R2. Based on the frequency changes along the prophase, we propose that D1R1 (can) transit into D2R1 and, possibly via D1R2, back to D1R1. We favor two options for early D1R1 configurations: the first option involves recombinases loaded on one DSB end, where DMC1 is orientated at the 3' end and RAD51 more towards the 5' end, while the other end is bound by other ssDNA binding proteins (i). The second option is that D1R1 involves loading of recombinases on both ends as described for option 1, but those ends are in close proximity due to low-affinity homotypic interactions, and cannot be resolved (ii). In a subsequent stage, both ends have accumulated recombinases and the search for homolog starts whereby the ends undergo a conformation change (iii). DMC1 is loaded on the 3' ends and moves further away, while the two RAD51 clusters cannot be resolved, resulting in one large RAD51 nanofocus. When strand invasion takes place, first DMC1 is

removed from the 3' end and simultaneously a D-loop forms where single stranded binding protein will bind. This situation matches D1R2 (iv). When strand invasion is completed, the other end is still bound by recombinases, resulting in a D1R1 situation (v). Later, after second end capture, or during synthesis-dependent strand annealing, these recombinases will also be removed. This process is most likely not unidirectional as drawn here (for simplicity), and recombinases may be released and re-associate at various moments, leading to the formation of less frequent and more complex $D_xR_y$ configurations, until intermediates are stable for a long enough period to complete the process.

configuration has an asymmetrical overall structure (small far DMC1 and a large close DMC1 nanofocus) may argue against this, or may represent (already) partial loss of DMC1 from an invading strand.

As strand invasion occurs and the ssDNA binds the homologous DNA, the recombinases will be gradually removed and simultaneously, a D-loop will form. Once all DMC1 has been removed from one end of the DSB, the D2R1 may revert to a D1R2 configuration, if the two RAD51 entities become separated (Fig 10B[iv]). Also, other intermediate structures with more complex configurations cannot be excluded. Simultaneously, the ssDNA of the D-loop will be bound by ssDNA binding proteins (RPA, MEIOB, SPATA22). When the recombinases have been completely removed from the invading strand, a D1R1 configuration may become prominent again (Fig 10B[v]). After second-end capture, or when synthesis-dependent strand annealing occurs, eventually, also the recombinases from the other ssDNA end will be removed.

With this speculative model, we present a plausible relationship between the most frequently observed $D_xR_y$ configurations and corresponding loading pattern of recombinases on the ssDNA. These schematic drawings obviously are simplified representations of recombination intermediates that likely exist in variable configurations and three-dimensional shapes in vivo. We would also like to emphasize the possible role of chance in the number of initiated loading events at each DSB end, and also in the 3' or 5' localization of either recombinase, as well as the possibility that loading and removal may be dynamic processes. Also, configurations that do not contribute to the homology search and/or progression of repair may be unstable and therefore have a shorter lifetime. We also want to note that in this model we excluded the complex configurations with more than 3 nanofoci (like D3R1). Since these configurations are quite rare, we hypothesize that they represent rare recombination intermediates which for example correspond to intermediates that are in transition from one (more frequently occurring) intermediate into the next.

## Recombination intermediate changes in absence of HORMAD1

In absence of HORMAD1 two important changes in recombination intermediates were observed. First, we observed changes in the frequencies of the most prominent configurations that are in accordance with a possible repair-inhibitory function of HORMAD1: in *Hormad1*[−/−] leptotene, more D2R1 and less D1R1 are present than in wild type. Thus, it seems reasonable to assume that *Hormad1*[−/−] leptotene-like spermatocytes contain relatively more late repair intermediates, suggesting that the lifetime of the early recombinase intermediates is reduced or that the recombinase intermediates are turned over faster. This fits with the results of a study in *Hormad1*[−/−]*Spo11*[−/−] spermatocytes where exogenous induced DSBs were repaired faster [49]. Our data now provide the first indication that HORMAD1 may similarly affect processing of endogenous SPO11-induced DSBs. The change in lifetime of recombinase intermediates in HORMAD1-deficient mice may be linked to its proposed function in the interhomolog bias in such a way that intersister repair would be expected to occur faster (the sister is nearby, so the search for homology may take less time) than interhomolog repair. However, it may also be that in absence of HORMAD1 both interhomolog as well as intersister repair occurs faster, since we cannot discriminate between recombinase accumulations at sites that

will be repaired by either the homolog or the sister. Our analyses of the sex chromosomes, that must repair via intersister repair in the nonhomologous regions does indicate a higher frequency of D2R1 configurations, but this is most clearly observed in the PAR, where repair using the homologous chromosome appears a more likely event.

The second key observation in *Hormad1*[-/-] spermatocytes, is the overall reduction in size of recombinase nanofoci. This is an unexpected observation, and one possible explanation is that it could be caused by reduced resection. Recent results have shown a minor role for EXO1 and a major role for ATM together with an unknown helicase in mediating end resection in mouse meiosis [64–66]. However, up to now there is nothing known about any involvement of HORMAD1 in this process. A perhaps more likely explanation for the reduction in recombinase nanofoci size could be faster turnover of the recombinases, but smaller recombinase nanofoci may also reflect a different (3D) folding structure of the filaments. Finally, less recombinase protein loading in favor of binding of other ssDNA binding proteins may also result in smaller recombinase nanofoci. This issue may be resolved once we are capable of "counting" molecules through the expression of directly labeled proteins, suitable for this type of analysis, which would then also allow live cell analysis to study dynamics of protein turnover at DSB sites.

## A new function for HORMAD1 as inhibitor of SC coiling

Super-resolution microscopy is emerging as highly suitable tool for studying the SC [31,67,68]. Using dSTORM, we observed that HORMAD1 localized as two parallel axes along axial, as well as lateral elements (Fig 1). This is in accordance with the study of Xu et al. (2019) [52], who used dSTORM in combination with expansion microscopy to unravel the localization of cohesins and SC components. Based on an axial view of the SC, they suggested a core-shell-like model, where SYCP3 represents the core and HORMAD1 represents the shell [52]. However, based on our 3D-STORM analysis, we favor a parallel-axes model where HORMAD1 forms a double axial structure, one along each side of the SYCP3 axis. In addition, based on other studies with higher resolution than dSTORM, SYCP3 may also form a double axial structure, not resolved in our images [67,69].

In addition to the visualization of the SC in wild type spermatocytes, we also investigated the SC in the absence of HORMAD1 with dSTORM (Fig 9). This led to the observation of an unexpected phenotype of this meiotic mutant. Organization of the SC is adapted in *Hormad1*[-/-] in such a way that coiling of the complex is increased. The helical structure of the SC was first observed in various organisms in the 70's using electron microscopy [57,70]. To the best of our knowledge, HORMAD1 is the first protein reported to be involved in suppressing coiling of the SCs in mice. Coiling of the SC expresses itself in synapsed SC regions, and is dependent on HORMAD1. Strangely enough, HORMAD1 is mainly present at unsynapsed regions, which suggest that the axial elements themselves may already be coiled and when two axial elements approach each other, this results in formation of the double-helical structure. Comparable to two twisted wires approaching each other resulting in double twisted wire. Similar to the double-helix of DNA, it is plausible that the structure of the axial elements determines the degree of torsion and that HORMAD1 plays a crucial role in determining the stiffness of the axes. Our data also shows that the presence of SYCP1 promotes coiling of the SC, and this has also been reported for the desynaptic (*dy*) mutant in maize [71]. Finally, we have also revealed a small but clear decrease in SC coiling in the absence of SPO11. This indicates an inhibitory role for meiotic recombination sites in coiling. However, the number of recombination sites is of course more severely affected in the *Spo11*[-/-] than in the *Hormad1*[-/-], while the effect on coiling is much stronger in the *Hormad1*[-/-]. To integrate these findings we speculate that there is a certain balance in recombination-dependent coiling inhibition, and SYCP1-dependent coiling

stimulation, the latter dampened by HORMAD1. In absence of HORMAD1, there will be less recombination-dependent coiling inhibition (because there are fewer DSBs formed), and the coiling stimulating effect of SYCP1 would be enhanced. In the SPO11 mutant, only the recombination-dependent coiling inhibition would be lost, resulting in a milder effect.

In the past, several hypotheses about the functional relevance of SC coils have been put forward. The first hypothesis suggested that the SC coils could represent the place of recombination nodules, but this has been refuted, using detailed EM analyses in rat spermatocytes [57]. Another hypothesis posited a relation between coiling and partner choice in homologous recombination, but so far there is no evidence to support this [72–74]. Our results now suggest that coiling is regulated by a complex interplay between at least two coiling inhibiting (HORMAD1, SPO11) and one coiling promoting (SYCP1) factor. Since HORMAD1 is also involved in promoting overall synapsis [34], it might be that suppression of coiling by HORMAD1 could somehow help to achieve proper functional synapsis, potentially via regulation of binding sites of SYCP1.

## Concluding remarks

Studying both RAD51 and DMC1 as well as SC proteins at high resolution allowed us to interpret recombination intermediates in relation to localization and function of HORMAD1. Our super-resolution dual dSTORM approach revealed a hitherto undiscovered phenotype of the SC in *Hormad1*$^{-/-}$ spermatocytes, where the SC has a more coiled structure than SCs in wild type spermatocytes. We propose that HORMAD1 modulates stiffness and/or the helicity/gyration of axial elements which prevents the SC from coiling (too) frequently, and an additional role for meiotic recombination in the inhibition of coiling.

Additionally, our single repair focus analysis of three proteins involved in meiotic DNA repair allowed a detailed description of both RAD51 and DMC1 in combination with SYCP3 or HORMAD1. Absence of HORMAD1 yielded similar configurations of RAD51 and DMC1, but the lifetime of the early recombination intermediates was altered in leptotene, which support a role for HORMAD1 in the timing of repair. Also, the reduction in size of RAD51 and DMC1 nanofoci indicates an additional function in either end resection, recombinase filament size, filament structure, or recombinase turnover. Of the two preferred distances (~300 nm and ~900 nm) in the distribution of distances between two nearest nanofoci, the ~900 nm peak is lost in *Hormad1*$^{-/-}$ spermatocytes which suggest that two DSBs are generally around ~900 nm apart in wild type and that each $D_xR_y$ configuration represents one DSB. We propose a model for the configuration of recombinase intermediates in which RAD51 and DMC1 loading on the ssDNA is related to the configurations and progression of the meiotic prophase whereby we favor the transition of D1R1 into a D2R1 and back into a D1R1 as repair proceeds. Additional research will be required to further unravel the molecular details of meiotic recombination at single DSB sites. Such research should make use of 3D super-resolution imaging, combined with visualization of DNA and/or other single-stranded DNA binding proteins.

## Material and methods

### Ethics statement

All procedures were in accordance with the European guidelines for the care and use of laboratory animals (Council Directive 86/6009/EEC). All animal experiments were approved by the local animal experiments committee DEC (Dutch abbreviation: Dier Experimenten Commissie) Consult and animals were maintained under supervision of the Animal Welfare Officer.

## Mice

Three wild type and three *Hormad1* knockout mice (C57BL/6 background) [34] were used for the three-color dSTORM experiments and two wild type mice were used for the two-color dSTORM experiments. For the STED experiments and control stainings we used mutant mice which were previously described: *Sycp1*[-/-] [75], *Sycp3*[-/-] [76], *Hormad2*[-/-] [77], *Spo11*[-/-] [78], and *Dmc1*[-/-] [10] all of these are on C57BL/6 background, except *Dmc1*[-/-] and *Spo11*[-/-] which were on a mixed background (C57BL/6 x 129/Sv). Mice were socially housed in IVC-cages with food and water ad libitum, in 12-h light and dark cycles.

## Meiotic spread preparations and immunofluorescence

Nuclei of testicular cells were spread as previously described [79]. For dSTORM testicular cells were spread on 1.5 thickness round coverslips with high precision (170±5 μm) coated with 0.01% poly-L-lysine (Sigma, P8920).

Immunofluorescence staining on spread preparations was performed as described before [49]. In brief, after washing with PBS, the spreads were blocked and incubated overnight at room temperature with primary antibody. After washing and blocking, the secondary antibody was incubated for two hours at room temperature. Finally, for STED microscopy the slides were embedded in Prolong Gold. For dSTORM the slides were stored in PBS.

## Antibodies

For primary antibodies, we used previously generated guinea pig anti-HORMAD1 (1:100, [41]), anti-SYCP3 (1:200, gift from R. Benavente, [80]) and rabbit polyclonal anti-RAD51 (1:1000, [81]), anti-SYCP1 (1:2500, [82]) and mouse monoclonal anti-DMC1 (1:1000, Abcam, ab11054), anti-SYCP3(1:200, Abcam, ab97672), For secondary antibodies, we used goat anti-guinea pig IgG Alexa 647 (1:500, Abcam, ab150187), goat anti-rabbit IgG Alexa 647 (1:250, Invitrogen, A21245), goat anti-rabbit IgG CF568 (1:500, Sigma Aldrich, SAB4600310), goat anti-mouse IgG CF568 (1:500, Biotium, 20109) and goat anti-mouse IgG Atto 488 (1:250, Rockland, 610-152-121S). For STED microscopy we used a goat-anti mouse IgG Alexa 555 (1:500, Invitrogen, A-21422). Control immunostainings in *Dmc1*[-/-] and *Spo11*[-/-] spermatocytes, which were imaged using a Zeiss Confocal Laser Scanning Microscope 700 with a 63x objective immersed in oil (images were taken with the same intensity), confirmed loss of DMC1 signal, and loss of signal of both recombinases respectively (S9 Fig).

## dSTORM imaging

dSTORM imaging was performed as described before [31] with minor adaptations. In brief, the coverslips with immunofluorescence staining were placed in a Attofluor Cell Chamber (ThermoFisher, A7816). To perform dSTORM imaging, the following buffer was used: 0.56 mg/ml glucose oxidase (Sigma, G2133), 34 μg/ml catalase (Sigma, C9322), 25mM MEA (Sigma, m6500) and 20% glucose in Tris-HCl pH8 with 10 mM NaCl. dSTORM imaging data were acquired on a Zeiss Elyra PS1 system using a 100x 1.46NA oil immersion objective. The TIRF angle was determined manually for each channel whereby the penetration depth was set between 190–200 nm. The data was acquired using a 512x512 Andor Ixon DU 897 EMCCD camera. For a super-resolution image, 12000 frames were taken with an acquisition time of 33 ms. High laser power used for dSTORM imaging can bleach fluorophores of higher wavelengths. Consequently, imaging was performed from high to low wavelength, starting with Alexa 647 to detect HORMAD1/SYCP3 (or HORMAD1 in two-color dSTORM). This is

followed by CF568 to detect RAD51 (or SYCP3/SYCP1 in two-color dSTORM) and finally Atto 488 to detect DMC1.

## dSTORM image analysis

Localizations were drift corrected (model-based approach) and grouped (max on time = 50 frames, off gap = 2 frames, capture radius = 4 pixels) using ZEN2012 software (Carl Zeiss). We used the fluorescent 100 nm fiducials (Thermo Fisher, T7279) for alignment of the different channels where we used the Alexa 647 channel as base. The alignment was performed in the Fiji platform [83] using a custom Fiji Plugin based on the DoM Plugin (www.github.com/ekatrukha/DoM_Utrecht). Localization data was analyzed and processed using the R-package SMoLR [84]. Images of the whole cell were processed using a Gaussian distribution with a pixel size of 5nm. For dSTORM experiments involving one channel (*Sycp3*⁻/⁻ spermatocytes stained for SYCP1), the images were generated using ZEN2012 software.

## HORMAD1 localization analysis

For HORMAD1 localization analysis dSTORM was performed on immunostainings of SYCP1 and SYCP3 in combination with HORMAD1. Intensity profiles of chromosomal axes were generated using the SMolR-generated [84] super-resolution images. Lines were placed perpendicular and spaced by 25 nm on a manually drawn line (length varied between 1–2.5 μm) through the center of the chromosomal axes. Of each perpendicular line an intensity profile was calculated using the function 'getProfile' by Fiji [83]. Using R, the average of all intensity profiles was calculated to generate an average intensity profile of the selected axis region. In total, for both synapsed and unsynapsed regions, 8 intensity profiles from 3 different nuclei were averaged and centered, resulting in one average intensity profile.

To generate a random control of HORMAD1 localization along the axial elements, an unsynapsed region of a chromosomal axis (shown in Fig 1I) was chosen. After measuring the intensity profile as described above, we first determined the boundary of the localization events, to create a mask. Next, we took the number of localization events that were within this mask, to generate a new random pattern with random precisions per localization event in the range of the measured localizations. We generated 50 such random patterns within this mask. The random intensity plot that is shown in Fig 1I is the average of all the 50 line profiles of the 50 generated random localization event patterns.

## 3D-dSTORM imaging and analysis

3D-dSTORM was performed as regular dSTORM but with the use of an astigmatic lens. Localization data was processed with Huygens Localizer version 21.04 (Scientific Volume Imaging, The Netherlands, http://svi.nl). Beads were used for z-alignment, z-drift and channel alignment. Fiji was used to optimize the visualization of 3D-images.

## STED microscopy

STED microscopy was performed on spermatocyte spreads of various mutant mice immunostained for primary anti-SYCP3 antibody and secondary antibody coupled to an Alexa 555 fluorophore. Images were acquired on a Leica SP8 tau STED equipped with an 86x/1.20 water immersion objective using a pulsed white light laser set to 550 nm and a continuous wave 660 nm depletion laser (notch filter applied). The emission signal was recorded between 560 and 610 nm and signal time was gated between 1 and 7 ns. Samples were imaged with a line accumulation of 16 to get sufficient signal. The pixel size for the STED images was 25–30 nm. To

improve the resolution, the lens was corrected for coverslip thickness using a motorized cover-slip correction collar for every slide. For every mutant 10–20 nuclei were imaged (pooled of two animals). In Fiji, a line was drawn manually through the center of synapsed regions to measure the length. Number of twists were counted manually. The average distance between twists was calculated by dividing the length of the chromosomal axis by the number of twists plus one per nucleus.

Because SYCP3 staining was not possible in the *Sycp3^{-/-}* mouse model, we performed dSTORM on SYCP1 (as described above). Similar to the HORMAD1 localization, we measured intensity profiles of three regions per nuclei (7 nuclei in total of both wild type and knock out) and combined them to an average intensity profile.

### Recombination foci analysis

Recombination foci were automatically selected from the dSTORM image in Fiji as follows. Step 1: SYCP3/HORMAD1 signal was used to create a mask of the chromosomal axes by applying a Gaussian Blur (sigma = 10 nm). Step 2: An automatic signal level threshold Huang dark was applied, followed by the Fiji function "Analyze Particles" with a threshold setting of 50000 nm$^2$ and a circularity from 0–0.89 to exclude beads. Step 3: This mask of the chromosomal axes was dilated by 125 nm to be able to include recombination foci that partially overlap with SYCP3. Steps 4a and 4b: For both RAD51 and DMC1 individually, a mask was also created after first setting an automatic signal level threshold (Huang dark), followed by application of the Fiji function "Analyze Particles" with a minimum size setting of 12500 nm$^2$ and 7500 nm$^2$ for RAD51 and DMC1, respectively. Steps 5a and 5b: These masks were dilated by 15 nm to merge small and close-by concentrations of localization events. Step 6a: A mask containing only regions positive for both RAD51 and DMC1 was generated. Step 6b: The masks of RAD51 and DMC1 were merged. Step 7: The axes mask was adjusted to include protruding recombinase foci by combining the axis mask (step 3) with the "double-positive" mask (step 6a). In this manner, recombination foci that still did not fully fit in the dilated mask of step 3 could be included, and the double-positivity helps to include only true foci that are localized near the SC. Step 8: The merged RAD51/DMC1 mask (containing all RAD51 and all DMC1 signal) from step 6b was overlayed with the final axis mask from step 7 and signal outside the axes mask was removed. Step 9: Recombination foci within the chromosomal axes mask were selected using the Fiji function "Find Maxima". Circular regions with a diameter of 750 nm were drawn around the centers to select a region of interest (ROI) covering all relevant RAD51 and DMC1 signal within such a region. Step 10: A single merging round was performed, in which ROIs with a center that was within 400 nm of each other were clustered as one to merge close by ROIs. Step 11: A manual adaptation was performed to remove beads and overlapping ROIs, to add missed ROIs or to recenter ROIs (these adjustments never involved more than ~10% of total DSB foci). See S10 Fig for step-by-step demonstration of this procedure. Please note that this selection defines the position of the ROIs, and that all localizations within these regions were used for further analysis.

Further analysis of the ROIs was performed in R [85]. Single molecule localization data of each ROI was used to fit a KDE function and thereby generate a binary image and configuration features as described before [31]. In brief, using R-package SMoLR, localization data was transformed to a binary image using a KDE function. Using a threshold (0.05 (axes components) or 0.15 (recombinases) localizations per nm$^2$) we were able to define clusters or so-called nanofoci of RAD51 and DMC1. Nanofoci smaller than 50 pixels (= 1250 nm$^2$) were excluded for configuration analysis. Besides image generation, SMoLR also provided shape features of nanoclusters (center of mass, area, localization event, angle, etc.). Based on the

number of nanofoci of DMC1 and RAD51, we assigned a configuration $D_xR_y$. ROIs with at least one nanofocus $> 500$ pixels (= 12500 $nm^2$) were included to minimize the inclusion of small, noise localizations. Moreover, we decided to focus on ROIs containing at least one cluster of each protein to reduce background signal because in most cases both proteins colocalize (S5 Fig). Details about the ROIs can be found in S1 Table.

Additional measurements were performed on the three main configurations–D1R1, D2R1 and D1R2. In case of D2R1 and D1R2, the two nanofoci of the same protein were classified as far and close based on the distance to the other protein. The rotation analysis was performed as described before [31]. In brief, (close) DMC1 was used as center and the other localizations were rotated so that either (close) RAD51 or (far) DMC1 were aligned above the close DMC1 center. For D1R2, the reverse was performed using close RAD51 as center. SMoLR and ggplot2 (Geom_density2d) were used to visualize the pooled localizations.

The distance analysis was performed as described before with small adaptations [31]. A line was drawn manually through the SYCP3/HORMAD1 signal at the axial (unsynapsed regions) and the two lateral elements (synapsed regions). Next, these lines were used for the distance measurements where the shortest distances from the nanofoci to the axes was calculated. Distances larger than 500 nm (1.9% of the data) were excluded from analysis. To decide if nanofoci were inside or outside the lateral elements, the direction of the center of mass of the nanofocus to the two synapsed lines were measured. If the direction was opposite, the nanofocus was inside the lateral elements, and in other cases the nanofocus was outside the lateral elements.

For the nearest neighbor analysis the center of mass of each nanofocus was used to determine distance to the closest nanofocus within a nucleus. To mimic the lower number of DSB in the *Hormad1$^{-/-}$*, two subsets of the wild type were chosen. The first subset was based on the number of nanofoci where we picked randomly 40% of the nanofoci within a nucleus. For the ROI-based subset we picked randomly 38.5% of the ROIs within a nucleus. The numbers were based on the ratio nanofoci or ROIs per cell for wild type towards knockout. For the nearest neighbor analysis where the axes were taken into account, the following strategy was used. First, the coordinate on the manually drawn axis line closest to the nanofocus was determined for all nanofoci. Next, the nearest neighbor of these coordinates was examined and the distance between the pair (along the manually drawn lines) was calculated. The manually drawn lines (for details see section about the distance to the axes) follow the SYCP3 signal, where a distinction was made between unsynapsed and synapsed regions. For synapsed regions, a single line between the two SYCP3 axes was drawn. We excluded single ROIs on an axes fragment, since there are only intra-ROI nanofoci distances in such an instance. We also excluded leptotene and leptotene-like nuclei from this analysis.

The angle between the DSB and SYCP3 in D2R1 was based on two linear fitted lines through the centers of the nanofoci (DSB) and the manually-drawn SYCP3 lines coordinates.

Plots were generated using ggplot2.

## Statistics

For the generation of the SYCP3/HORMAD1-RAD51-DMC1 dataset for wild type and *Hormad1$^{-/-}$* nuclei of 3 different animals of each genotype were pooled. Statistical analysis were performed in R [85]. The appropriate test and the population on which the test was performed (nuclei, ROI or nanofocus) is indicated in the text, legend, or S2 Table. * = $p < 0.05$, ** = $p < 0.01$, *** = $p < 0.001$.

## Supporting information

**S1 Fig. Individual intensity profiles of unsynapsed regions of Fig 1D.** Boxed region indicates which region was used to generate the intensity profiles. SYCP3 (red) and HORMAD1 (green). Scale bar represent 500 nm.
(EPS)

**S2 Fig. Individual intensity profiles of synapsed regions of Fig 1H.** Boxed region indicates which region was used to generate the intensity profiles. SYCP1 (red) and HORMAD1 (green). Scale bar represent 500 nm.
(EPS)

**S3 Fig. Nearest neighbor analysis with synaptonemal complex taken into account.** Histogram of relative frequency distribution of nearest neighbor distances between subsequent coordinates along the manually drawn axes that correspond to the intersection of the line from the center of mass from each nanofocus to the nearest SC or axial element. The analysis was performed for wild type DMC1 (top left) and RAD51 (bottom left), $Hormad1^{-/-}$ DMC1 (top right) and RAD51 (bottom right). Single ROIs on an axis fragment were excluded from the analyses, and leptotene(-like) nuclei were not included because the axes fragments at this stage are too small for a meaningful analysis. DMC1 is indicated in green and RAD51 in red. Distances were binned in 100 nm bins, distances larger than 3.5 μm were labelled as rest.
(EPS)

**S4 Fig. RAD51 and DMC1 configurations including configurations consisting of only one recombinase.** A) Barplot of fraction of ROIs with the indicated configuration of RAD51 and DMC1 nanofoci for wild type (n = 55 nuclei) and $Hormad1^{-/-}$ (n = 36 nuclei). Barplot of fraction of D1R0 and D0R1 per cell nucleus per substage for wild type (B) and $Hormad1^{-/-}$ (C). Error bars indicate standard deviation. p-values can be found in S2 Table.
(EPS)

**S5 Fig. Boxplot of area of nanofoci within configurations D1R1, D2R1, and D1R2 per stage in wild type and $Hormad1^{-/-}$ spermatocytes.** Leptotene (l), early zygotene (ez), mid zygotene (mz), late zygotene (lz), early pachytene (ep), leptotene-like (l-like), zygotene-like (z-like), early pachytene-like (ep-like). p-values can be found in S2 Table.
(EPS)

**S6 Fig. Boxplot of eccentricity of nanofoci within configurations D1R1, D2R1, and D1R2 per stage in wild type and $Hormad1^{-/-}$ spermatocytes.** Leptotene (l), early zygotene (ez), mid zygotene (mz), late zygotene (lz), early pachytene (ep), leptotene-like (l-like), zygotene-like (z-like), early pachytene-like (ep-like). p-values can be found in S2 Table.
(EPS)

**S7 Fig. Additional analyses for RAD51/DMC1 in relation to HORMAD1 and SYCP3.** A) Rotation analysis of all D1R1, D2R1 and D1R2 foci relative to HORMAD1 channel of wild type. From left to right: Images were rotated as indicated in the schematic drawing whereby the anchor (*) indicates the center and the goal (o) was rotated until it was aligned with the center. Summed kernel density estimation image of all D1R1, D2R1 or D1R2 foci with (close) RAD51 (red) and (close) DMC1 (green) combined with far-nanofocus (white) or HORMAD1 (magenta). Heatmap-style density plot of HORMAD1 localizations. B) Boxplot of percentage of RAD51/DMC1 colocalizing with axes for wild type (SYCP3 and HORMAD1) and $Hormad1^{-/-}$. Barplots of fraction of ROIs with an indicated configuration of RAD51 and DMC1 nanofoci for wild type (C) and $Hormad1^{-/-}$ (D) divided in unsynapsed and synapsed regions.

E) Example of calculation of the angle between the axes and RAD51/DMC1. F) Boxplot of angle between DSB and SYCP3 in D2R1 for both wild type and *Hormad1*[-/-]. Error bars in C) and D) indicate standard deviation. p-values can be found in S2 Table. n in A) indicates the number of ROIs. Scale bars represent 100 nm.
(EPS)

**S8 Fig. Coiling of the SC.** A) Plot of number of coils per chromosome averaged per nucleus. Median is indicated in red. B) Dotplot showing the relation between the number of coils per chromosome axes and the length of the chromosome axes. C) dSTORM and D) close-up of boxed region shown in C) image of wild type mid-zygotene nucleus immunostained for SYCP1 (white). E) dSTORM and F) close-up of boxed region shown in E) image of *Sycp3*[-/-] pachytene-like nucleus of immunostained for SYCP1 (white). G) Average intensity-profiles of SYCP1 at synapsed chromosomal axes for both wild type and *Sycp3*[-/-] (each consist of 21 profiles (3 regions per nuclei and 7 nuclei in total) for both genotypes). Scale bar represents 5 µm (C,E) and 1 µm (D,F).
(TIF)

**S9 Fig. Antibody specificity.** Confocal images of wild type, *Spo11*[-/-] and *Dmc1*[-/-] zygotene (-like) nuclei immunostained for SYCP3 (red) and RAD51 (green) (left) and SYCP3 (red) and DMC1 (green). Also images with a higher contrast are shown for each recombinase staining. Scale bars represent 5 µm.
(TIF)

**S10 Fig. Step-by-step explanation of semi-automatic ROI selection.** Details can be found in the material and methods.
(TIF)

**S1 Table. This Excel file contains details of the ROIs which were analyzed in the wild type and *Hormad1*[-/-] nuclei.**
(XLSX)

**S2 Table. This Excel file contains all the statistical details relating to Fig 2, Fig 3, Fig 4, Fig 5, Fig 7, Fig 8, Fig 9, S4 Fig, S5 Fig, S6 Fig, S7 Fig.**
(XLSX)

## Acknowledgments

We would like to thank Ricardo Benavente (University of Würzburg, Germany) for kindly sharing the SYCP3 antibody and all the members of the Erasmus MC Developmental Biology department and Optical Image Center for useful discussions.

## Author Contributions

**Conceptualization:** Lieke Koornneef, Willy M. Baarends.

**Formal analysis:** Lieke Koornneef, Johan A. Slotman, Esther Sleddens-Linkels, Marco Barchi.

**Funding acquisition:** Joost Gribnau.

**Investigation:** Lieke Koornneef, Johan A. Slotman, Esther Sleddens-Linkels, Marco Barchi.

**Methodology:** Lieke Koornneef, Johan A. Slotman, Wiggert A. van Cappellen.

**Supervision:** Joost Gribnau, Adriaan B. Houtsmuller, Willy M. Baarends.

**Visualization:** Lieke Koornneef.

**Writing – original draft:** Lieke Koornneef, Willy M. Baarends.

**Writing – review & editing:** Lieke Koornneef, Johan A. Slotman, Esther Sleddens-Linkels, Wiggert A. van Cappellen, Marco Barchi, Attila Tóth, Joost Gribnau, Adriaan B. Houtsmuller, Willy M. Baarends.

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
