## [Decision Letter · Decision Letter 0]

17 Mar 2022

Dear Willy,

Thank you very much for submitting your Research Article entitled 'Multi-color dSTORM microscopy in Hormad1-/- spermatocytes reveals alterations in meiotic recombination intermediates and synaptonemal complex structure' to PLOS Genetics.

The manuscript was fully evaluated at the editorial level and by independent peer reviewers. The reviewers appreciated the attention to an important topic but identified some concerns that we ask you address in a revised manuscript. Many of the concerns may be addressed by text editing, but reviewers 2 and 3 have suggested some minor additional experiments that could strengthen your observations.

We therefore ask you to modify the manuscript according to the review recommendations. Your revisions should address the specific points made by each reviewer.

[LINK]

Yours sincerely,

Paula E. Cohen

Associate Editor

PLOS Genetics

Gregory P. Copenhaver

Editor-in-Chief

PLOS Genetics

Reviewer's Responses to Questions

**Comments to the Authors:**

Reviewer #1: This interesting manuscript reports on a study of early meiotic DNA double-strand break recognition and repair making use of super-resolution 3-color dSTORM microscopy. The authors study both WT (wild type) and Hormad1-mutant spermatocytes. The findings allow the authors to propose models for RAD51 and DMC1 recombinase configurations that may ultimately provide insight into the mechanisms of early DSB recognition. Moreover, they provide evidence that the HORMAD1 protein may inhibit formation of coils in the SC.

Overall, the manuscript is a lucid and well-written explanation of the experimental strategy and data, accessible to those interested in meiosis who may not be familiar with the imaging technology. The Introduction is generally an excellent presentation of background and strategy. It could be enhanced by adding a more compelling rationale for the selection of the Hormad1 mutant, and more clearly stating the precise questions that the research answers (e.g., what new will be resolved by the application of 3-color dSTORM, why was it not possible before, and how will the Hormad1 mutant contribute?). The references in this section, many of which are labeled as “recent,” are, in reality, rather old; the authors should cite some of the more recent reviews on meiotic DSB formation and repair.

The experimental analyses are quite clearly presented and enhanced by figures that contain both gorgeous images of the fine structure of meiotic axes and clear summary intensity distribution plots. However, one notable absent feature in imaging is the chromatin, which also is not designated in the final model figures. While it is realized that the investigators were studying DSB recognition and repair complexes, an introductory image showing chromatin background would be useful. Likewise, the addition of chromatin loops to the final models would help readers to set the overall chromosomal context, so important in consideration of homolog bias, etc.

As the authors acknowledge, the absence of Hormad1 results in very subtle differences in spacing and size of DxRy foci on the axes – what have we learned from these small differences about either HORMAD1 protein function or assembly of repair complexes? The unexpected finding that HORMAD1 protein might negatively impact SC coiling is quite interesting. The analysis of DxRy complexes in the PAR is a great addition to this paper, but what about the Hormad1 mutant in this respect? And what about coiling along the X and Y and in the PAR? Given these considerations, it is especially important that the authors more fully justify the use of Hormad1 mutants and provide a more compelling summary of precisely what has been learned from its use.

In summary this is an interesting manuscript that gives us a tantalizing taste of the power of 3-color dSTORM microscopy in elucidating the mechanisms of meiotic DSB recognition and repair. We all eagerly await the time, hopefully not too far in the future, when this kind of “nano microscopy” converges with equally revealing molecular analyses. Because the authors have already published valuable papers in this realm (refs 29 and 31), the overall impact of this contribution will be enhanced by clear statements of what is newly found and how it might change our view of recombination complexes and/or function of HORMAD1 protein. This is briefly stated in the Abstract (lines 30-32) but deserving of greater clarity and attention.

Reviewer #2: In their manuscript, Koornneef et al use three-color dSTORM to analyze the positional patterning of the recombinases, RAD51 and DMC1, in reference to axial elements in WT and Hormad1-/- mutants. Their analysis, when synthesized with prior observations from ChIP-seq data, allows the authors to provide a fascinating, albeit highly speculative, model for DMC1/RAD51 arrangement and dynamics from 3’-end resection through strand invasion during meiotic recombination. This, in combination with the truly stunning dSTORM imaging, makes this manuscript an engaging and thought-provoking read. I particularly appreciated the analysis of DMC1/RAD51 positioning on the XY (although this could have been bolstered with the use of FISH to identify the XY in all nuclei, and I wish the authors spent more time discussing the complex DMC1/RAD51 arrangements, i.e., D3R1, D3R2, etc.). I also appreciate the use of multiple mutants (Spo11, Sycp1, Hormad2) in the evaluation of HORMAD1’s role in SC coiling. I have several reservations about this study, however, that prevent me from giving it an unequivocally glowing review:

1) The inclusion three-color dSTORM as well as the Hormad1-/- mutants allows the authors to address some important follow-up questions from their recent 2020 paper, but many of their results – particularly distance between nanofoci between WT and mutants and the positioning of DMC1/RAD51 in respect to SYCP3/HORMAD1 - seem to be very incremental developments; whether they significantly advance beyond the scope of the 2020 paper is debatable. Further, the observation that HORMAD1 appears to inhibit coiling of the SC is fairly shallow and feels like an afterthought compared to the 8 figures devoted to teasing out the fine, detailed measurements of DMC1 and RAD51 nanofoci positional data. And the differences observed between Hormad1-/- mutants and WT are so subtle, making it difficult to appreciate the biological significance of observations (e.g., marginally increased focus size in mutants). It is unclear what the broader significance of this study is (given very similar observations in the 2020 paper and decades of conventional IF staining).

2) Many of the initial observations of HORMAD1 localization and DMC1/RAD51 ROI numbers are very concerning. First, the observation of HORMAD1 on synapsed regions is contrary to the generally observed trend of HORMAD1 absence along synapsed regions. Second, the number of RAD51/DMC1 ROIs observed in leptonema – zygonema (~150) seems a little low, while the number observed in pachynema (~50) is strikingly high! In C57BL/6J adult males, zygotene RAD51 foci numbers average around 250 while pachytene RAD51 foci are all but unobserved. The observation of ~50 RAD51/DMC1 ROIs in WTs is very concerning.

3) I am also concerned that, because most other studies rarely observe RAD51 and DMC1 in WT pachytene spermatocytes, the authors’ observations of DMC1/RAD51 nanofoci positions and dynamics in pachytene may not be illustrative of “typical” meiotic DSB repair (i.e., via interhomolog of invasion in advance of either of the two CO pathways or SDSA).

4) Although there is a notable decrease in the frequency of D1R1 from leptotene/early zygotene to mid-/late-zygotene (and a comparable increase in the frequency of D2R1 in the same time frame), the magnitude of these changes (particularly the latter) is ultimately very subtle – the authors’ speculation that there is a progressive evolution from D1R1 -> D2R1 -> D1R1 needs further empirical support. This proposed evolution also seems to be contradicted by the observation that D1R1 is less common in later prophase.

5) The authors’ analysis of D2R1 orientation around the SC showed a minority of RAD51 inside of lateral elements; however, the authors conclude that this supports a model for a D2R1 dumbbell structure with RAD51 located in the center of the SC – this is just not born out by the majority of the data.

6) I want to stress that I appreciate the amount of work the authors put into image acquisition and analysis. Although the total number of ROIs analyzed was high, the number of biological replicates was shockingly low – only three animals analyzed per genotype, with approximately 10 nuclei or fewer analyzed per prophase stage per animal.

7) Distinctions of what constitutes a single focus are a little nebulous. For instance, D2R1 configurations have one DMC1 focus much smaller than the other, with an elongated/oblong RAD51 focus. Is it possible the oblong “focus” is actually two foci that are too close to be resolved by dSTORM? With this in mind, it is difficult to discern what variation in focus size and shape mean in term of number of whether the focus is “real” or nonspecific/background staining.

8) It is critical throughout the manuscript to be clear about what stage is being analyzed – at several points throughout I was uncertain whether the authors were discussing leptotene or early-, mid- or late-zygotene. Similarly, it is critical the authors specify what mouse strain the WT and mutants are on.

Reviewer #3: This manuscript by Koornneef et al. reported dSTORM imaging of RAD51, DMC1 and axial/lateral elements (SYCP3) in mouse spermatocytes. By comparing wildtype and Hormad1-/- knockout mice, the authors detected some minor changes of DMC1-RAD51 configurations in the absence of HORMAD1. Integrating recent findings using RAD51 and DMC1 ChIP-seq in mice, authors proposed a revised model to interpret microscopic images (some modifications compared to their previous model) and accommodated major DxRy configurations into recombination pathway.

In general, the results shown here provide new insights into how RAD51 and DMC1 are possibly organized at DSB sites. Yet, it is a bit disappointing that these results cannot explain HORMAD1 functions nor directly understand more about how HORMAD1 affects DMC1-RAD51 patterns and SC coiling.

Having said that, this is an interesting story. My main concern is that the revised model to explain DMC1-RAD51 configuration is partially based on comparisons of distances/distribution/numbers of ROI and nanofoci in wild type and Hormad1-/- knockout mice. However, without knowing the distribution of decreased DSBs in mutants, whether HORMAD1 is involved in repair pathway choice and whether HORMAD1 affects DSB end processing, it seems to be shaky to use their patterns of DMC1-RAD51 to explain DMC1-RAD51 configurations in wild type. This concern, along with other more minor comments and suggestions for the manuscript are listed below.

1. Abstract is not well summarized. Seemingly, the study aims to understand how recombinases are loaded on ssDNA in the absence of HORMAD1. However, I think results have little information that can link DMC1-RAD51 patterns to HORMAD1 functions.

2. I think the most important results of this study can be the modification of previous interpretation of DMC1-RAD51 configurations on each end of DSBs, but it may require supporting evidence of other mutants that are directly defective in DSB processing or repair.

3. Line 109-110 states that Hormad1 -/- leads to meiotic arrest at zygotene. But later, analysis used pachytene-like stages for mutants. Perhaps, adding more description to clarify how staging was performed for mutant and/or rephrasing “meiotic arrest at zygotene”.

4. Lines 122-126 and also lines 527-531, studies mentioned here were performed in spo11-deficient backgrounds, where DSBs are exogenously induced and ssDNA processing and repairing may not be canonical, so it may be inappropriate to use these findings to explain DMC1-RAD51 patterns observed in Hormad1-/-.

5. Lines 148-151, please describe the analysis/sampling of randomized control in Material and Methods.

6. Lines 155-156 state that two HORMAD1 rods run along each side of the axial elements, and also in the same plane with SYCP1. To support this, please show a representative “3D-dSTORM” of HORMAD1 and SYCP1 in frontal and axial views.

7. Please show original images if acquired by widefield microscopy, as the Fig 1A has traces of background cropping.

8. Please provide supporting data or references to demonstrate specificity of RAD51 and DMC1 antibodies used in this study.

9. Lines 168-170, for this study and conclusion made later, selection of ROIs is critical. Please provide a supplemental figure to demonstrate procedures step-by-step of recombination foci analysis for lines 711-727.

10. Since size of ROIs may change how the authors interpret DMC1-RAD51 configurations, can authors track axes or SCs and measure distances of associated DMC1-RAD51 on each axis/SC? It should confirm the 800-900 nm distance. Also it will be helpful to count how many associated DMC1-RAD51 configurations in wild type and mutants to support ROIs being correctly assigned.

11. The authors can consider presenting distribution of nearest neighbor analysis of nanofoci first. Along the analysis of tracing axes or SCs, these results may justify parameters of ROI selection. So, the notion that number of ROIs in wild type and mutants are similar to their DSB numbers is more convincing. It also can support the proposed model in which each configuration represents one DSB (both ends of a DSB), rather than one end of a DSB.

12. Please show an additional real image (together with SYCP3 signals) for Fig 2O.

13. Need better description for Fig 2O, such as double lines vs single lines.

14. Lines 197-200, if the reduction of DSB numbers in Hormad1-/- knockout spermatocytes is not uniform, the lower frequency in 750-1050 nm may not reflect reduced DSB numbers in mutants. So, I suggest additional analysis to track axes or SCs in wild type and mutants at early stage.

15. Lines 270-273 and Fig 4C. In addition to the eccentricity, analyzing “Feret” may provide significant difference their shapes.

16. In Fig 7A, why the overlying axis form a circle? Did authors only cut ROIs for analyses? From the schematic drawing, does it mean that axis of D1R1 configuration is parallel to axis?

17. Line 377 and Fig 7H, nanofoci outside of lateral elements include signals in between lateral elements (inside of the SC)? I assume authors refer to signals outside of SC?

18. Which configuration is most likely to form CO? D2R1 or D1R2? Or is there any, even more complex configurations that may possibly represent CO designation?

19. The D1R2 number seems to be consistent across stages, are they similar to CO numbers per meiosis in mouse?

20. Please deposit original images in public data repository.

**Have all data underlying the figures and results presented in the manuscript been provided?**

Reviewer #1: Yes

Reviewer #2: Yes

Reviewer #3: Yes

PLOS authors have the option to publish the peer review history of their article (what does this mean?). If published, this will include your full peer review and any attached files.

Reviewer #1: No

Reviewer #2: No

Reviewer #3: No

---

## [Decision Letter · Decision Letter 1]

15 Jun 2022

Dear Willy,

We are pleased to inform you that your manuscript entitled "Multi-color dSTORM microscopy in Hormad1-/- spermatocytes reveals alterations in meiotic recombination intermediates and synaptonemal complex structure" has been editorially accepted for publication in PLOS Genetics. Congratulations!

Yours sincerely,

Paula E. Cohen

Associate Editor

PLOS Genetics

Gregory P. Copenhaver

Editor-in-Chief

PLOS Genetics

Comments from the reviewers (if applicable):

Reviewer's Responses to Questions

**Comments to the Authors:**

Reviewer #2: This is a fascinating work that provides valuable insight into RAD51 and DMC1 patterning from 3’-end resection through strand invasion of meiotic prophase I. Through gorgeous dSTORM imaging, this work has probed nuanced questions about the role of HORMAD1 in influencing the duration of early DSB repair and the temporal dynamics of repair intermediates. Moreover, the work provides a springboard for exciting new mechanistic interrogation of how HORMAD1 affects DMC1/RAD51 patterning and SC coiling. Much of my initial issues with the manuscript centered around a lack of clarity in staging of cells and methodological details around the evaluation of ROIs. I thank the authors for their thorough and prudent response. I am thoroughly satisfied with the changes the authors have made to the manuscript and I feel their alterations have considerably improved the clarity of the work.

Reviewer #3: This manuscript has improved a lot after revision. The authors explained and responded to all reviewers' comments/questions. I have only two suggestions for improving Figures to help readers.

1. For the merged images of rotated configurations with SYCP3 (magenta), close RAD51 (red), and close DMC1 (green) in Figures 7A and S7, the RAD51 signals are invisible. It is difficult to follow text descriptions if RAD51 is not seen in the rotated images.

2. For Fig 10, I have to say that the Bii is very strange, and does not make much sense to me. If authors think two ends of a single DSB are too close to resolve, perhaps, drawing an overlapping DMC1/RAD51 on two ends of a DSB may be easier to understand.

**Have all data underlying the figures and results presented in the manuscript been provided?**

Reviewer #2: Yes

Reviewer #3: Yes

PLOS authors have the option to publish the peer review history of their article (what does this mean?). If published, this will include your full peer review and any attached files.

Reviewer #2: No

Reviewer #3: No

**Data Deposition**

http://datadryad.org/submit?journalID=pgenetics&manu=PGENETICS-D-22-00082R1

**Press Queries**

---

## [Editor Report · Acceptance letter]

12 Jul 2022

PGENETICS-D-22-00082R1 

Multi-color dSTORM microscopy in *Hormad1-/-* spermatocytes reveals alterations in meiotic recombination intermediates and synaptonemal complex structure 

Dear Dr Baarends, 

We are pleased to inform you that your manuscript entitled "Multi-color dSTORM microscopy in *Hormad1-/-* spermatocytes reveals alterations in meiotic recombination intermediates and synaptonemal complex structure" has been formally accepted for publication in PLOS Genetics! Your manuscript is now with our production department and you will be notified of the publication date in due course.

With kind regards,

Marianna Bach

PLOS Genetics

On behalf of:
